# Spatial transcriptomics using multiplexed deterministic barcoding in tissue

Johannes Wirth [1], Nina Huber[1], Kelvin Yin[1], Sophie Brood[1], Simon Chang[1], Celia P. Martinez-Jimenez [1,2] & Matthias Meier [1,3]

Spatially resolved transcriptomics of tissue sections enables advances in fundamental and applied biomedical research. Here, we present Multiplexed Deterministic Barcoding in Tissue (xDBiT) to acquire spatially resolved transcriptomes of nine tissue sections in parallel. New microfluidic chips were developed to spatially encode mRNAs over a total tissue area of 1.17 cm$^2$ with a 50 μm resolution. Optimization of the biochemical protocol increased read and gene counts per spot by one order of magnitude compared to previous reports. Furthermore, the introduction of alignment markers allowed seamless registration of images and spatial transcriptomic spots. Together with technological advances, we provide an open-source computational pipeline to prepare raw sequencing data for downstream analysis. The functionality of xDBiT was demonstrated by acquiring 16 spatially resolved transcriptomic datasets from five different murine organs, including the cerebellum, liver, kidney, spleen, and heart. Factor analysis and deconvolution of spatial transcriptomes allowed for in-depth characterization of the murine kidney.

Single-cell transcriptomics (scT) has revolutionized the concept of cellular heterogeneity and led to the development of comprehensive reference maps of cells typically isolated from biopsies, tissues, and whole organisms[1–4]. These methods elucidate cell-to-cell communication, and tissue architecture, which play key roles in tissue homeostasis, tissue repair, and disease progression. However, tissue dissociation protocols cause loss of spatial information and alteration of cell type proportions, removing critical information to understand cellular crosstalk and the microenvironment.

To overcome this limitation, spatial transcriptomics (ST) has been developed based on imaging, sequencing, or a combination of both methodologies[5]. Imaging-based methods exploit in situ hybridization probes to detect single transcripts with high spatial resolution down to the subcellular scale; however, the need for targeted probes limits the study to a predetermined set of genes[6,7]. Instead, for sequencing-based techniques, RNAs are barcoded with DNA molecules to encode the spatial position; enabling, untargeted detection of mRNAs of the whole transcriptome. While commercially available technology like Visium Spatial resolves tissue spots with diameters on the order of tens of microns, recent technical improvements made in high-definition spatial transcriptomics (HDST)[8] and STEREO-seq[9] resolve transcripts down to sub-micron spot sizes. Alternative methods, including sci-SPACE[10] and XYZeq[11], barcode cells within the tissue before retrieval. In the next step, nuclei or whole cells are isolated from the tissue, and their transcriptomes are sequenced together with the positional barcodes.

Depending on the research question, several parameters, such as spatial resolution, detection limit, screening area, accessibility, compatibility with existing workflows, and costs, are weighed against each other to select the most suitable method for an experiment. For instance, high-resolution methods either require specialized equipment to manufacture the components and establish the analysis in the lab, or are proprietary, which leads to higher costs. Lower resolutions are, in turn, associated with the loss of single-cell resolution because of the larger resolved spot sizes. This, however, can be compensated for

[1]Helmholtz Pioneer Campus, Helmholtz Munich, Munich, Germany. [2]TUM School of Medicine, Technical University of Munich, Munich, Germany. [3]Center for Biotechnology and Biomedicine, University of Leipzig, Leipzig, Germany. ✉e-mail: celia.martinez@helmholtz-muenchen.de; matthias.meier@helmholtz-muenchen.de

by computationally integrating scT with ST and inferring the cell type composition of each spot[12–15].

Deterministic barcoding in tissue (DBiT-seq) is a cost-effective and openly accessible platform to scale ST[16]. DBiT-seq uses microfluidic channels to barcode tissue sections using DNA oligonucleotides and allows the integration of multi-omics information, including antibodies[17], epigenomics[18], and chromatin accessibility readouts[19].

In this study, we present Multiplexed Deterministic Barcoding in Tissue (xDBiT), a method for acquiring spatially resolved transcriptomes from nine fixed tissue sections in parallel. Optimization of the chemical protocol and workflow of the DBiT-seq method led to an increase in transcript reads and gene counts per $50 \times 50\ \mu m$ spot. The introduction of alignment marks onto the tissue sections enabled the seamless acquisition of transcriptomic reads and spatial registration with high-resolution images. Together with technological advances, we provide an open-source computational pipeline to transform the raw sequencing data from an xDBiT experiment into Scanpy-compatible data file formats[20,21].

To demonstrate the functionality of xDBiT, we acquired spatially resolved transcriptomic datasets of 16 tissue sections from five different murine organs, including the cerebellum, liver, kidney, spleen, and heart. Using the kidney as model tissue, we show that xDBiT can be used in conjunction with factor analysis to perform an in-depth characterization of organs and identify spatially patterned genes. Finally, we demonstrated that xDBiT can resolve rare cell types upon cell-type deconvolution using scT data, allowing cost-efficient research projects on spatiotemporal expression dynamics in longitudinal studies and multi-organ comparisons.

## Results

### Multiplexed Deterministic Barcoding in Tissue (xDBiT)

To enable multiplexing, increase sequencing depth, and improve the image data quality of the DBiT-seq methodology, we developed a Multiplexed Deterministic Barcoding in Tissue (xDBiT) workflow (Fig. 1A and Supp. Fig. 1A-G). For an xDBiT experiment, nine fresh frozen tissue sections with a maximum area of $0.4 \times 0.4$ mm were positioned in a $3 \times 3$ grid layout on a glass substrate (Supp. Fig. 1H). Tissue sections were fixed with PFA and the nuclei, cytoskeleton, and selected proteins were stained using a standard immunofluorescence protocol (see Methods). High-resolution images were acquired before the xDBiT run to obtain high-quality images without the introduction of artifacts from the downstream ST processing steps. Subsequently,

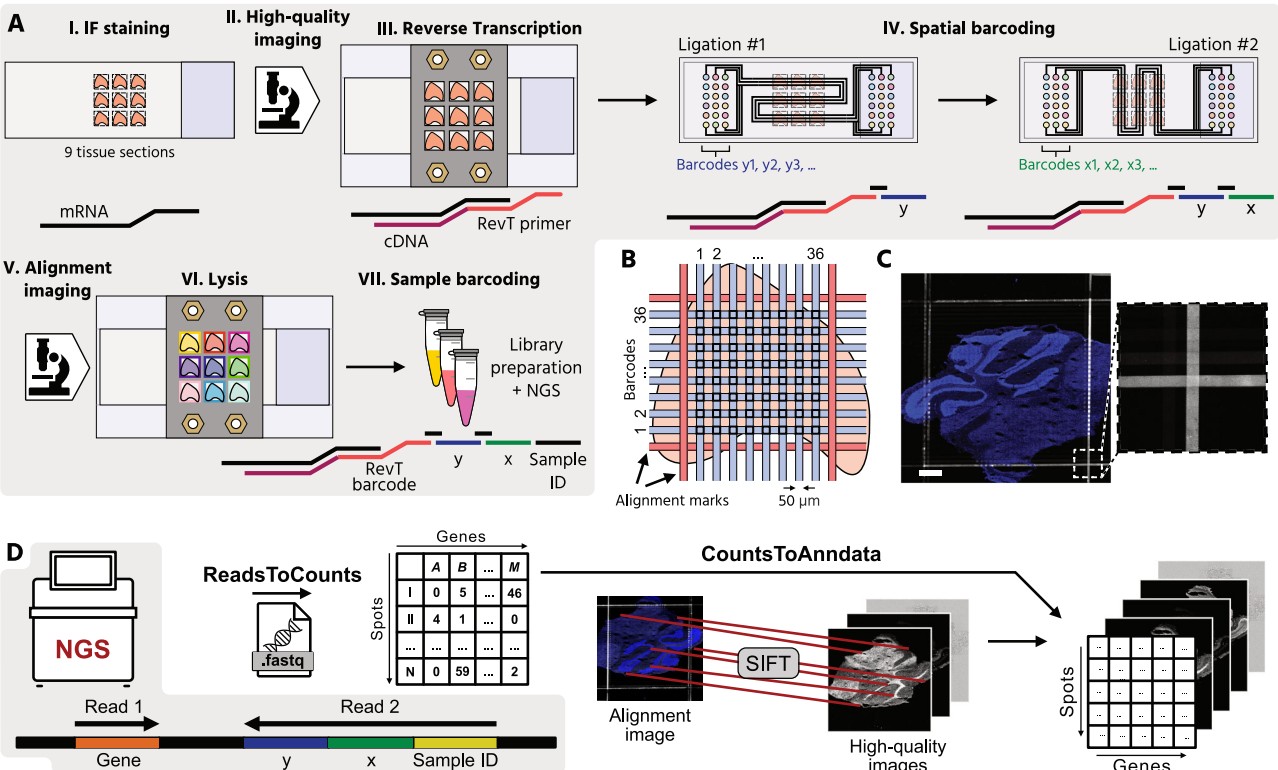

**Fig. 1 | Multiplexed deterministic barcoding in tissue (xDBiT). A** Workflow of xDBiT: (I) Deposition of PFA-fixed cryosections with a thickness of 10 μm onto a glass substrate in a 3 × 3 matrix layout and IF staining. (II) Acquisition of high-quality fluorescence images. (III) Reverse transcription of mRNA into cDNA in all tissue sections with the help of a 3D-printed 9-well adapter. (IV) Spatial barcoding of the cDNA in two sequential ligation steps. Each ligation step is performed with an individual PDMS chip. The two chips were designed with an orthogonal serpentine channel layout to generate a 38 × 38 spot array on the tissue sections. The channel intersections generate a spatially encoded spots with an area of 50 μm × 50 μm and a center-to-center spot distance of 100 μm. Alignment marks to register IF images and spatial transcriptomic coordinates are generated by filling the outermost channels with a fluorescently labeled anti-BSA antibody. The antibody targets the blocking reagent BSA, which was deposited on the tissue surface during IF staining. (V) Re-imaging of the tissue section to obtain an alignment image. (VI) Lysis of the nine individual tissues is achieved using the 3D printed 9-well adapter. (VII) Each sample is individually indexed before sequencing. **B** Channel layout to generate alignment marks for the registration of high-resolution images and spatial transcriptomic spot coordinates. **C** Representative fluorescence images of the alignment marks with zoom-in image. Blue and white denote DAPI counterstain and anti-BSA staining, respectively. Similar results were obtained from the other eight capture areas and from two independently run xDBiT chips. Scale bar: 500 μm. **D** Schematic of the xDBiT computational pipeline. NGS reads are transformed into a spot-gene count matrix using the *ReadsToCounts* script. Alignment- and high-quality images are registered via their DAPI signal using the SIFT algorithm[22] and then aligned to the count matrix with the *CountsToAnndata* script. The resulting integrated datasets are compatible with *Scanpy* and *Squidpy* analysis pipelines[20,21,23]. PFA Paraformaldehyde, PDMS Polydimethylsiloxane, BSA Bovine serum albumin, IF Immunofluorescence, NGS Next-generation sequencing, SIFT Scale-invariant feature transform. Icons were created using Affinity Designer 2.

mRNAs within tissue sections were reverse transcribed using a 3D printed 9-well adapter (Supp. Figure 1F), which separated each section and reduced the reaction volume to 80 μL per sample. The reverse transcription (RevT) primer carried a hybridization site to ligate the spatial barcodes in the following working steps, and a poly(T) 3′-end to bind to and reverse transcribe all polyadenylated mRNAs (Supp. Fig. 2). In addition, the RevT primer contained a unique, 8-bp long sequence to barcode the samples during the RevT reaction (Supp. Figure 2). Analogous to DBiT-seq, spatial barcoding of the resulting cDNA was performed using two sequentially aligned polydimethylsiloxane (PDMS) chips. The first PDMS chip was clamped onto the tissue, creating 38 parallel, and horizontally aligned, microchannels (50 μm × 50 μm) on top of each tissue section, and allowing DNA barcodes to be flushed over the tissue (Supp. Figure 1I). The DNA barcodes were ligated to the cDNA within the underlying tissue and thereby encoding the positions of the horizontally directed channels.

The second PDMS chip resembles the first chip, with the difference that the 38 microchannels run vertically over the tissue section to barcode the cDNA in the tissue via ligation with an identifier for the vertically directed channel. The spatial barcoding resulted in a grid of 1444 uniquely barcoded spots, each with a width of 50 μm. In contrast to the original DBiT-seq approach, the microchannels were guided in serpentines over the glass substrate, which allowed us to address nine tissue sections in parallel and increased the scanning area from 25 to 116.64 mm² (4.66-fold increase). Importantly, we found that dehydration of the tissue sections with ethanol was essential to ensure the optimal attachment of the PDMS chips. To enable registration of the spatial transcriptomic spots to the image data, the two outermost channels were filled with an alignment marker solution (Fig. 1B) consisting of an anti-BSA antibody that binds to the BSA-blocked surface. After the second round of ligation, the tissue sections were imaged again to record alignment marks and stained nuclei (Fig. 1C). Finally, the 9-well adapter was attached to the slide to lyse the tissue sections individually. Within xDBiT, tissue multiplexing can be achieved after either the reverse transcription with barcoded primers or sample retrieval by indexed library preparation.

For the analysis of xDBiT spatial transcriptomic data, we developed a 2-step computational pipeline that integrates raw next generation sequencing (NGS) reads and image data (Fig. 1D). In the first step of the pipeline (*ReadsToCounts*), spatial coordinates and transcript information were extracted from the raw sequencing reads. Reads without valid x- or y-barcode were discarded. After genomic alignment, data were transformed into a spot/gene count matrix. In the second step (*CountsToAnndata*), the SIFT algorithm[22] was used to register the high-quality and alignment images based on their DAPI channels and calculate an affine transformation matrix. The transformation matrix was used to project the xDBiT spots onto the high-quality image to generate an integrated AnnData file compatible with Scanpy and Squidpy[21,23] for further analysis.

## xDBiT performance analysis

To demonstrate the performance improvements of xDBiT, we first acquired ST data from murine liver sections using the standard DBiT-seq protocol published by Liu et al. DNA read counts per spot for the liver samples were comparable to the read counts obtained with DBiT-seq on mouse embryo sections (Fig. 2A). The lower number of genes per spot for the liver sample (Fig. 2B) can be explained by the highly homogeneous cellular composition of the liver, which results in low cell type variation per spot. In the next step, we performed xDBiT using two sequentially improved protocols. In the first optimization round, we changed the chemical composition of the initial reactions of the DBiT-seq protocol, namely, the reverse transcription and spatial barcoding reactions. In comparison to DBiT-seq, the reverse transcription reaction, which generally suffers from low yields[24], was performed on

whole tissue sections in the 9-well adapter at a concentration of 10 U/μL rather than inside the microfluidic channels to increase the availability of the reverse transcriptase. Furthermore, the concentration of ligase was increased from 15 to 20 U/μL. Spatial barcoding was achieved by two sequential ligation steps, which were performed at lower temperatures and required shorter incubation times than RevT, thus reducing the risk of leakage between channels. Together, the chemistry optimization resulted in a three-fold increase in both read and gene counts per spot compared to DBiT-seq (Fig. 2A, B). In the second optimization round, we dehydrated and dried the tissue sections before applying each of the two PDMS chips to improve the attachment of the microfluidic channels. To fill microfluidic channels equally, inlet ports were primed with DNA barcode solutions by centrifugation and bubble traps were added at the transition of the inlets to the microchannels (Supp. Figure 1G). Collectively, these changes increased the read and gene counts per spot two-fold and four-fold, respectively (Fig. 2A, B).

It is noteworthy that the structural integrity of the cryo-sections was strongly reduced after the deterministic barcoding workflow because of the physical alignment of the PDMS chips to the tissue and the enzymatic treatments. Thus, to obtain high-quality image data, which are currently underutilized by standard ST methods[25], we acquired images before and after the xDBiT workflow. While the images before the xDBiT workflow exhibited high-quality features (Fig. 2C I), the features in the images collected after the deterministic barcoding steps showed lower quality (Fig. 2C II) but contained the marks required to align the ST data (Fig. 1C). Nuclei integrity was unaffected after the xDBiT workflow, and thus alignment images could be registered to high-quality images using the provided *CountsToAnndata* pipeline to transfer the positional information of the alignment marks to the high-quality images (see the Methods section).

To demonstrate the quality of the spatial transcriptomic data, we projected the raw sequencing read counts per spot onto the nuclei images as shown exemplarily for *Actb* in Supp. Fig. 3A. Resulting overlay images showed stripe artifacts consisting of rows or columns of spots with higher or lower read counts compared to their neighboring elements. These artifacts have been reported previously[16] and can be effectively removed by normalizing each spot by the total number of reads of the respective spot (Supp. Fig. 3B).

Sample multiplexing within the xDBiT approach was achieved by implementation of a serpentine channel design. For this, the microfluidic channels were elongated and the lengths of the resulting channels varied between 117.7 mm and 165.7 mm. We characterized the effect of the channel length on the fluid flow behavior on a PDMS chip by measuring the volumetric flow rate in all 38 channels when applying a constant vacuum of 300 mbar to the outlets. Flow rates showed a negative linear correlation with the channel length as it was expected from the Hagen-Poiseuille equation[26] (Fig. 2D). Between the shortest and the longest channels on the PDMS chips the flow rate differed by 26.5%. Time intervals for washing steps were adjusted to the lowest flow rate on the chip to ensure a minimal volume exchange of 15 μL per channel.

However, the advantage of sample multiplexing with xDBiT also carried the risk of cross-contamination between samples. To check for potential leakage between the individual wells of the 9-well adapter during the RevT reaction, food dye-colored aqueous solutions were used. Within an interval of 24 h no visible cross-contaminations were detected (Supp. Fig. 5E). Subsequently, potential cross-contaminations occurring after the RevT step were investigated in one xDBiT experiment with eight liver sections, leaving the center well of the 3 × 3 grid empty (Fig. 2E, Methods). Analysis of the resulting sequencing reads revealed that only 5.5 to 9.5% of the RevT barcodes were cross-contaminations from neighboring samples (Fig. 2E, Supp. Table 4, and Supp. Fig. 5A). Notably, from the empty well (Fig. 2E, sample B2) the cDNA concentration was not sufficient to perform a library

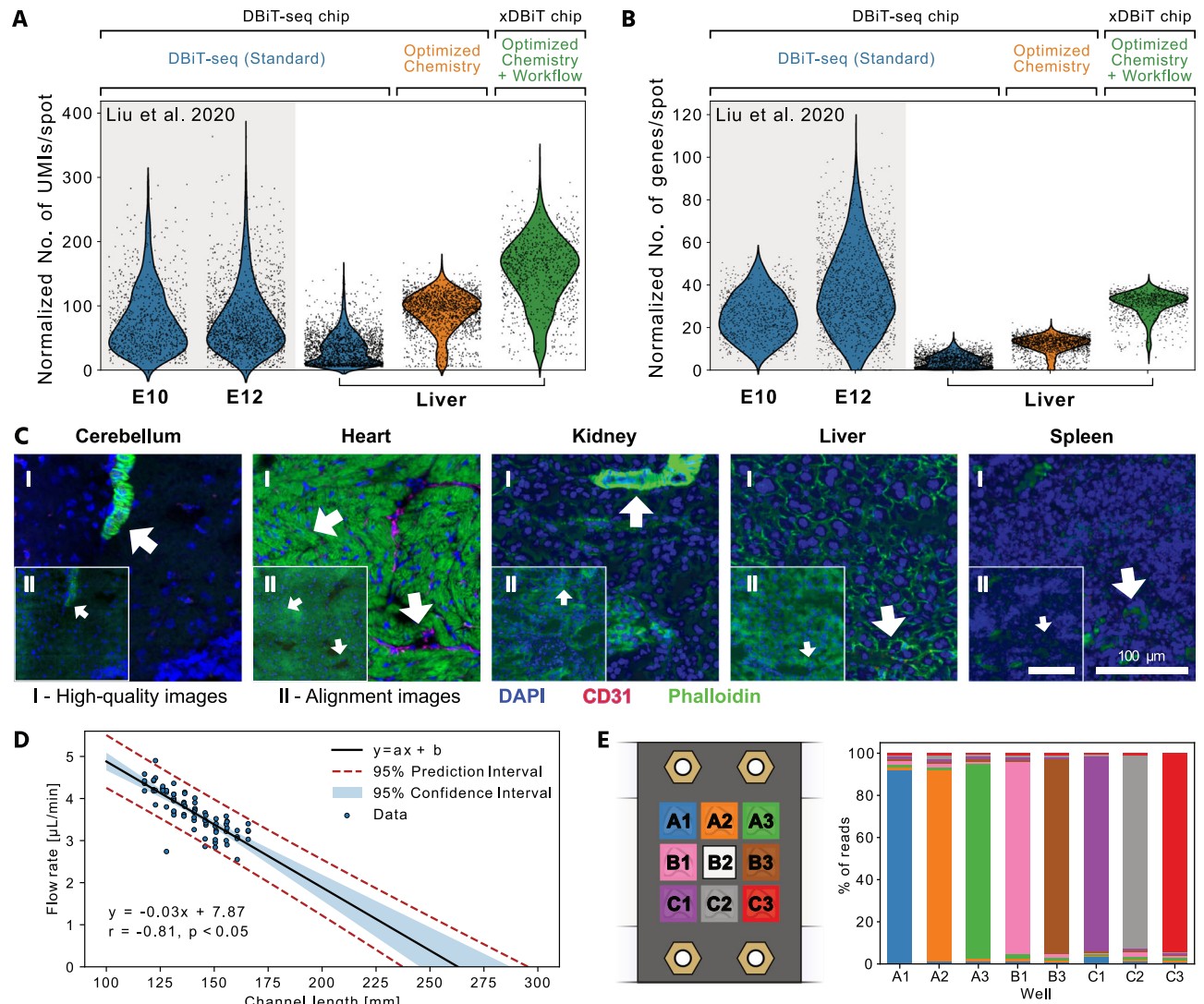

**Fig. 2 | Performance of xDBiT and quality assessment of the imaging data.**
**A**, **B** Violin plots showing the number of UMIs **A** and genes **B** per 50 μm × 50 μm spot, normalized to the sequencing depth (per $10^6$ reads) of the respective experiment. The xDBiT method was obtained by sequential optimization of the chemical protocol and workflow of the DBiT-seq method (Standard). For comparison, counts from the previously published DBiT-seq datasets from fresh frozen sections of murine embryos at stage E10 and E12 were added[16]. **C** Representative high-quality (I) and alignment images (II) of mouse tissue sections acquired during xDBiT workflow. Blue, magenta, and green colors denote DAPI, CD31, and phalloidin fluorescence signals, respectively. The quality of the fluorescence signal decreases during the xDBiT workflow, which demonstrated the need for the initial high-quality imaging round. Each organ has been measured between two and four times, showing similar results. For a detailed overview on the experiments see

Supp. Table 2. Scale bar: 100 μm. **D** Scatter plot showing the measured volumetric flow rate as a function of the microfluidic channel length. The range of measured channel lengths comprises all lengths integrated within the xDBiT PDMS chip design. A linear regression model was fit using the ordinary least squares method and the black line shows the resulting mean of the fit. The blue colored region and the red dotted line indicate the 95% confidence and prediction intervals, respectively. r Pearson correlation coefficient, p p-value. **E** Left: Experimental setup of the reverse transcription. Colors denote the barcode that was used for the respective well. Right: Stacked bar plot showing the percentage of reads found in a well, which carried a certain well barcode. Colors denote the barcode and match the experimental outline on the left. UMI Unique molecular identifier, PDMS Polydimethylsiloxane.

---

preparation for sequencing. Importantly, with the double barcoding strategy, cross-contaminations can be removed within the *ReadsToCounts* pipeline (see Methods).

Taken together, the xDBiT workflow provides a multiplexing method for ST and paired high-quality imaging. The cost per tissue section is on the order of 125€ (see Supp. Figure 4).

### Spatially resolved multi-organ dataset with xDBiT
To demonstrate the broad applicability of xDBiT, we generated 18 spatially resolved datasets from six different murine organs, including the kidney, heart, cerebellum, spleen, liver, and pancreas (Fig. 3). Depending on the organ, the UMIs and genes per xDBiT spot varied between 5000–20,000 and 1000–5000, respectively (Fig. 3A,

B). The pancreas samples showed low UMI and gene counts and were therefore excluded from further analysis (Supp. Fig. 5F). The sequencing depth for all organ samples was close to saturation, which was evaluated by computational subsampling analysis (Supp. Fig. 6). Samples were only barcoded by indexing primers during the library preparation. For removal of cross-contaminations, the background expression level of genes was measured based on ST spots without underlying tissue. Subsequently, only genes with an expression level higher than twice the standard deviation of the mean background signal were used for downstream analyses. Matching genes of the individual samples before and after background correction against the HOMER database[27] confirmed the depletion of cross-contamination signals (Supp. Fig. 5B, C).

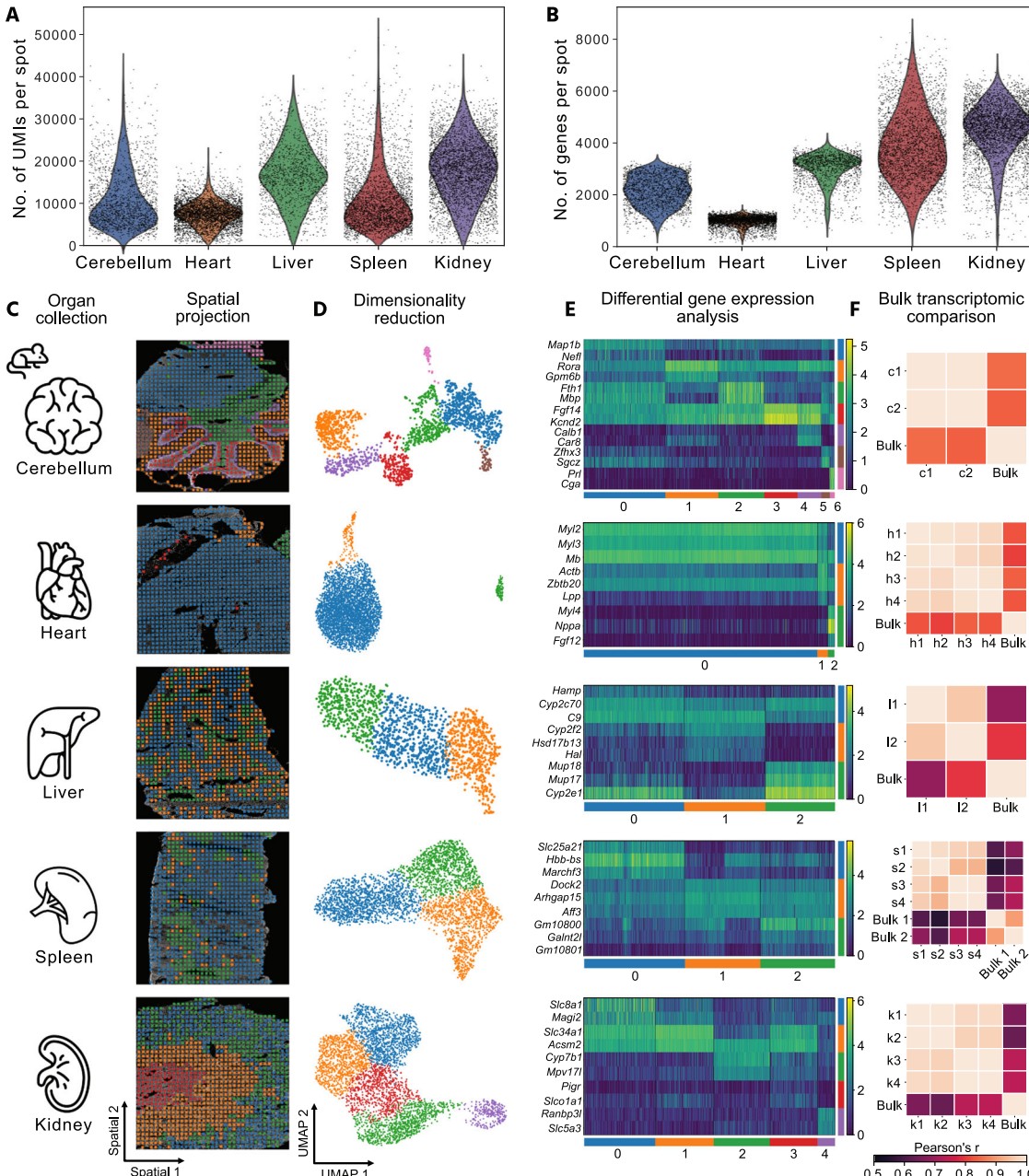

**Fig. 3 | Spatial transcriptomics of multiple murine organs with xDBiT. A, B** UMI and gene counts per spot obtained with the xDBiT method for tissue sections of different murine organs. **C** Representative DAPI images of five murine organs overlaid with xDBiT spatial gene expression. **D** Two-dimensional embedding of the xDBiT transcriptomic data using UMAP[28]. Each spot represents one xDBiT spot. Colors denote clusters that were determined using the Leiden algorithm[29]. **E** Top differentially expressed genes of the identified Leiden clusters within the corresponding organs. While the continuous color bar denotes the log-transformed gene expression level, the distinct color bars refer to the Leiden clusters. **F** Pearson correlation of tissue bulk transcriptomes from the ENCODE database[36, 37] with pseudo-bulk xDBiT spatial transcriptomic datasets of organ samples from two independent xDBiT experiments. Detailed information is provided in Supp. Table 2 and 3, respectively. UMI Unique molecular identifier, UMAP Uniform Manifold Approximation and Projection, c Cerebellum, h Heart, l Liver, s Spleen, k Kidney, numbers indicate replicates. Icons were created by Freepik, Tru3 Art and Smashicons from Flaticon.

After preprocessing and dimensionality reduction using Uniform Manifold Approximation and Projection (UMAP)[28], the data showed no visible batch effects (Supp. Fig. 7). Clustering using the Leiden algorithm[29] and projection of xDBiT spots onto the respective microscopy images displayed spatially distinct clusters (Fig. 3C, D). Further, differential gene expression (DGE) analysis between Leiden clusters revealed known marker genes for the substructures of the respective organs (Fig. 3E). For example, in the heart tissue section, we found the cardiomyocyte markers *Myl2*, *Myl3*, and *Mb* to be the top differentially expressed genes[30–32]. In the liver section, the zonation markers *Cyp2f2* and *Cyp1a2* were expressed in mutually exclusive areas[33] indicating

that transcriptomic XY resolution is sufficient to define zonated gene expression patterns. In the cerebellar sections of the brain, we were able to identify structures such as the arbor vitae (cluster 2) and the cerebellar cortex comprised of clusters 1, 3, and 4 (Fig. 3C). Cluster 4 delineated the course of Purkinje cells in the cerebellar cortex, as confirmed by gene ontology (GO) term enrichment analysis using the STRING algorithm[34] and the Brenda Tissue Ontology[35] (see Supp. Fig. 8A and Fig. 3).

In the spleen, DGE analysis revealed genes that are known to be expressed in the red pulp, such as *Slc25a21* or *Hbb-bs* for cluster 0, and genes expressed in the white pulp, such as *Arhgap15* and *Aff3* for

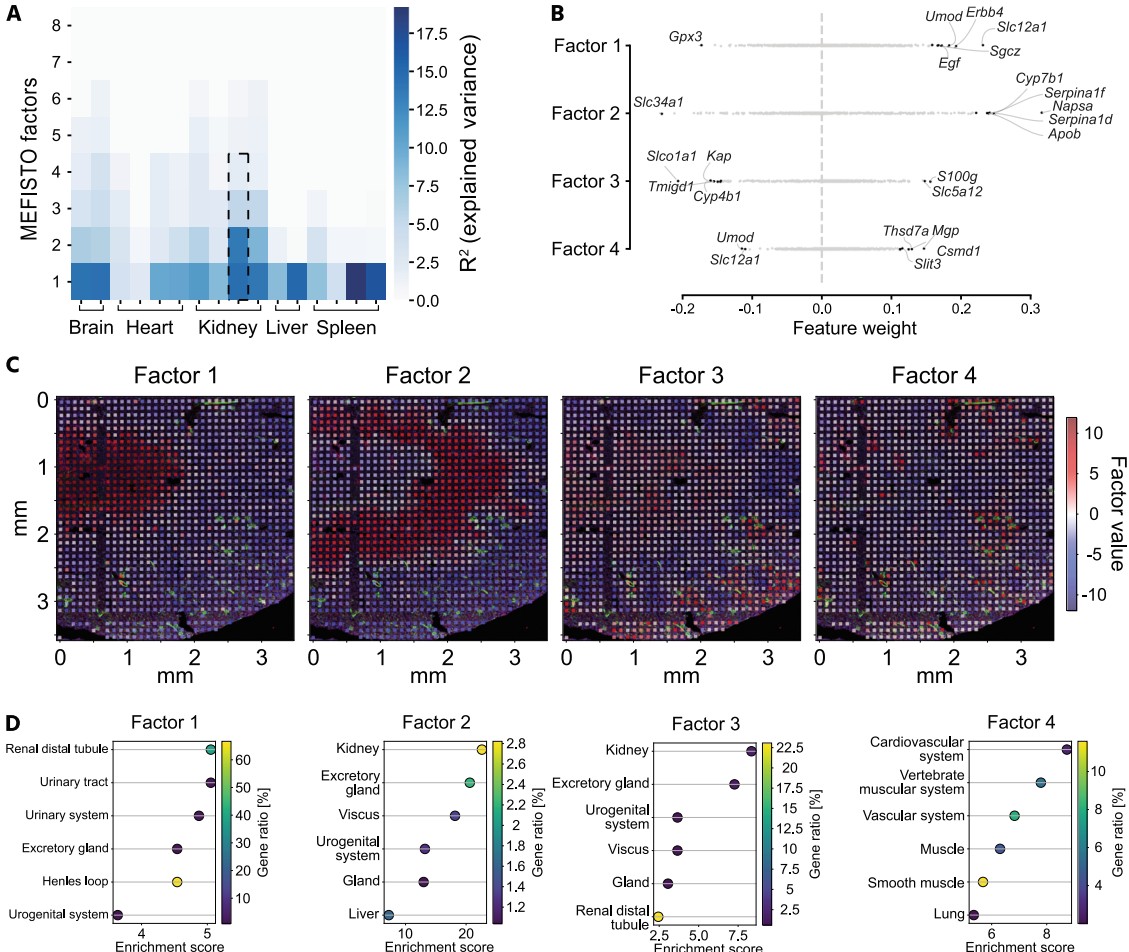

**Fig. 4 | Spatial pattern analysis on xDBiT spatial transcriptomic data using MEFISTO factor analysis. A** Results of MEFISTO factor analysis of all individually analyzed xDBiT datasets[38]. The heat map shows the percentage of explained variance ($R^2$) for the first 8 factors of each xDBiT replicate. Dotted frame indicates the representative kidney sample which was selected for downstream analysis in **B**–**D**. **B** Scatter plot showing the weight of all genes for the first four MEFISTO factors in the representative sample. Genes with the highest positive or negative weights are labeled. **C** High-resolution fluorescence images with overlaid xDBiT spots colored for the values of the respective first four MEFISTO factors. Blue: DAPI, Green: Phalloidin; Red: CD31. **D** GO term enrichment analysis using the STRING algorithm[34] and the *Brenda Tissue Ontology* database[35]. As input we used the top positively weighted genes (>95% confidence interval) of the first four factors to reveal functional and structural areas of the murine kidney.

cluster 2. GO term enrichment analysis confirmed the identity of tissue clusters (Supp. Fig. 8D). To further confirm the high quality of the xDBiT datasets, a pseudobulk xDBiT dataset was created and compared with published bulk RNA-seq datasets from the ENCODE project[36,37]. Pearson correlation coefficients between the xDBiT pseudobulk and bulk transcriptome data ranged from 0.55–0.83 (Fig. 3F).

### Characterization of spatial gene expression

For spatial gene expression pattern analysis of the xDBiT ST data, we applied MEFISTO, a factor analysis method to identify the driving sources of gene variation in high-dimensional datasets while accounting for spatial dependencies[38]. The factor analysis was performed separately for each tissue section and identified a set of previously unobserved variables, called factors. These factors reveal the covariance structure of the spatial transcriptomic dataset of the respective tissue section. Sections from the same organ showed a comparable number of factors that explained spatial gene expression variations (Fig. 4A). While tissue sections from structurally more complex organs like cerebellum or kidney contained up to six factors explaining the variance in gene expression, in homogenously structured organs like liver or spleen only two factors were sufficient. Investigation of the feature weights of individual factors revealed that the corresponding gene sets influenced the factors in a positive or

negative direction (Fig. 4B). To further evaluate the performance of MEFISTO on a structurally complex organ, kidney was chosen as model tissue and the first four factors of one kidney section were selected for downstream analysis (dotted frame in Fig. 4A). To show that MEFISTO captured structural areas within the tissue sections, we projected the factor values onto the fluorescence image of the respective kidney tissue section (Fig. 4C). Factors 1, 2, 3, and 4 define the anatomical regions of the inner and outer medulla, renal tubules in the cortex and medulla, and blood vessels in the kidney, respectively. Similarly, the spatial gene expression of the top positively weighted genes matched the patterns of their corresponding spatial factors (Supp. Fig. 9A). To support factor-to-region assignments, we performed GO term enrichment analysis with the top positively weighted genes of the first four factors (Fig. 4D). Analysis of factor 1 showed significant enrichment for terms related to Henle's loop, a functional structure of the kidney located in the inner medullary region. For Factor 2, the analysis did not show enrichment for specific anatomical regions, but positively weighted genes of this factor were cell type markers for proximal tubules, including *Napsa* and *Serpin1f*. Accordingly, the analysis of positively weighted genes of factor 3 showed significant enrichment in genes of the distal tubules located in the renal cortex. Lastly, the spatial pattern of factor 4 correlated with phalloidin and CD31 staining in the cortical and inner medullary regions of the kidney (Supp. Fig. 9C).

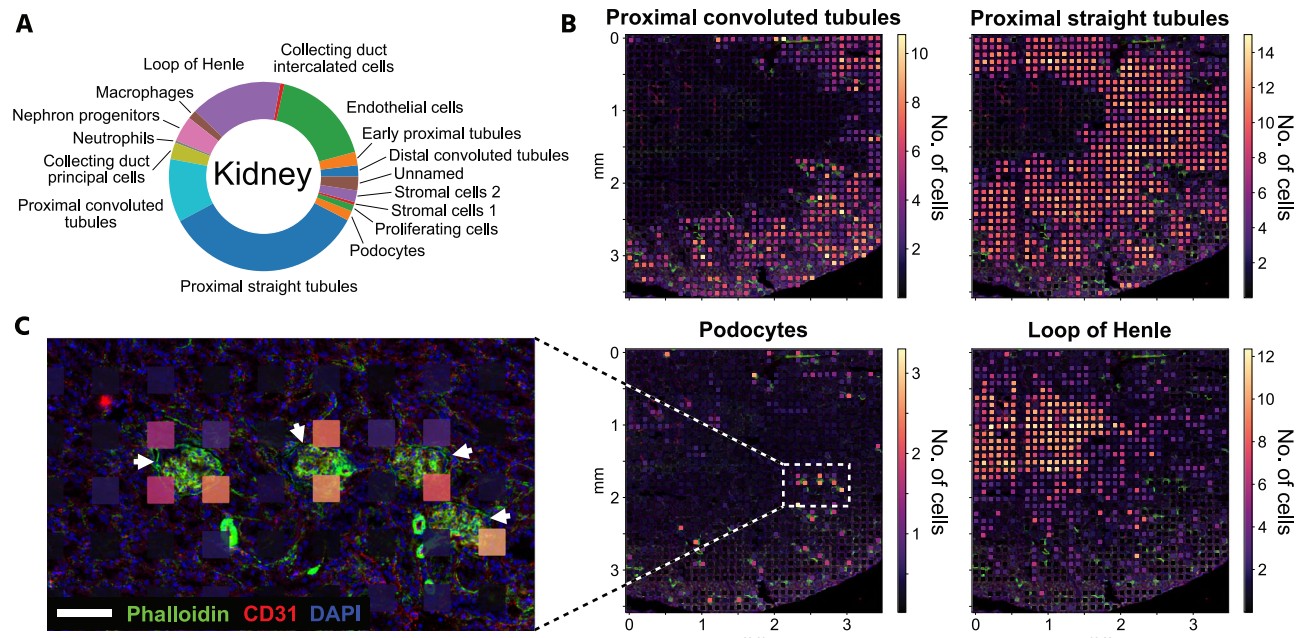

**Fig. 5 | Deconvolution of xDBiT kidney dataset to spatially resolve cell types.** **A** Data of one representative kidney section was deconvolved using *cell2location*[13] and a single-cell RNA-seq dataset of the murine kidney[41]. Pie chart shows the cell type composition of the deconvolved kidney section. **B** High-resolution fluorescence images overlaid with xDBiT spots colored for the deconvolution results of four representative kidney cell types. Spot colors correspond to the minimum number of cells predicted. **C** Detailed fluorescence image. Arrows denote the position of glomeruli. Overlaid xDBiT spots show the number of podocytes predicted by *cell2location*. DAPI (blue), phalloidin (green), and CD31 (red). Scale bar: 100 μm.

These findings were consistent with the GO term analysis, which showed that genes of the cardiovascular system were enriched. In conclusion, xDBiT ST data in combination with MEFISTO factor analysis allowed simultaneous identification and characterization of functional regions in tissue sections from multiple murine organs.

## Deconvolution of xDBiT kidney dataset to spatially map cell types

One challenging aspect of ST methodologies and their corresponding computational tools is achieving single-cell resolution across an entire tissue section. For example, existing spatial transcriptomic methods, including Visium Spatial[39], Slide-seqV2[40], DBiT-seq[16], and xDBiT, contain multiple cells per spot and are thus unable to reach single-cell resolution. However, single-cell information can be extracted from spatial transcriptomic spots with more than one cell using deconvolution methods[12–14]. In this study, the *cell2location* analysis tool[13] was used in conjunction with a published single-cell transcriptome dataset of the murine kidney[41] to obtain the cell-type compositions of each spot on an xDBiT kidney ST dataset (Supp. Fig. 10A). The most abundant cell types were cells from the proximal straight tubule (34.6%) and endothelial cells (17.3%), followed by cells from the loop of Henle (15.7%) and the proximal convoluted tubule (10.8%) (Fig. 5A). These findings are in agreement with those of previous studies that investigated the cell type composition of murine kidneys[42,43]. Furthermore, the predicted spatial distribution of these cell types matches the anatomical structure of the kidney[43] (Fig. 5B). This prediction was further validated by visual correlation of the inferred number of endothelial cells per spot and the fluorescence signal intensity of the endothelial marker CD31 in the kidney section (Supp. Fig. 10B). While cells of the proximal convoluted tubule were found predominantly in the cortex of the kidney, the number of cells of the proximal straight tubule was increased in the outer medulla. Cells of the loop of Henle were mainly predicted to be in the medullary region of the section, which coincides with the GO term analysis of MEFISTO factor 1 (see Fig. 4C, D). To further challenge the xDBiT dataset, we asked whether it is possible to map podocytes, which are cell types located within the

glomeruli and have a crucial role in renal filtering processes. High-quality fluorescence images allowed us to identify the position of glomeruli in the tissue section based on phalloidin staining of F-actin, which is a characteristic of glomeruli[44]. The number of inferred podocytes correlated well with the position of the glomeruli, showing high podocyte numbers in spots close to a glomerulus (Fig. 5C). That xDBiT spots did not fully align with the glomeruli suggests that the resolution of the spots was larger than the 50 μm × 50 μm area. This might be caused by the diffusion of molecules within the fixed tissue and beneath the microfluidic channels. Notably, podocytes are underrepresented in kidney datasets and require special isolation methods[45,46]. The proportion of podocytes detected solely by single-cell transcriptomic data was only 0.3%[41] whereas other, less biased studies predicted 3%, a much higher percentage of cells[42,47]. *Cell2location* inferred a podocyte proportion of 1.7% and thus a more realistic approximation of the kidney cell composition when ST was taken into account (Fig. 5A and Supp. Table 5). In summary, the use of xDBiT in conjunction with *cell2location* allows us to map all major renal cell types in a kidney section and generate a more accurate representation of rare cell types in complex microenvironments than scT alone.

## Discussion

Spatially resolved transcriptomes of tissues from multicellular organisms have greatly expanded our knowledge of complex cellular functions and cell-to-cell communication in healthy and diseased conditions. Single-cell transcriptomics, together with spatial transcriptomics, have become central technologies for mapping cell types in their tissue context and architecture. Most single-cell and spatial transcriptomic studies use a hypothesis-free and explorative design[25,30,48–50]. However, to pursue systematic and hypothesis-driven research approaches, ST technologies must comply with the increasing demand for providing multiple replicates per condition, time trajectories, or sampling multiple organs from the same individual at low costs.

In this study, we expanded the technology of *Deterministic Barcoding in Tissue* to simultaneously analyze nine individual tissue

sections. To achieve this, we developed new microfluidic chip platforms to spatially barcode mRNA transcripts in spots with an area of 50 μm × 50 μm. In combination with an optimized chemical workflow, the transcript number of reads and genes per spot were increased by 6 and 12-fold, respectively, compared to the original DBiT-seq method[16].

xDBiT allowed barcoding of the samples using the 9-well adapter during the initial RevT reaction and library preparation. Notably, we did not observe liquid exchange between the wells of the 9-well adapter. By barcoding samples both at the RevT and the sample indexing level, we demonstrated that less than 9% of the final sequencing reads resulted from cross-contaminations occurring after the RevT step, indicating low cross-contamination among samples. Importantly, within double-barcoded samples, cross-contamination signals can be removed computationally by selecting sequencing reads with matching RevT and sample indexing barcodes. Alternatively, when using only a single barcoding strategy, cross-contamination events can be removed computationally by analysis of the gene expression background. However, we recommend the double-barcoding of samples to exclude the possibility of sample cross-contaminations.

Further downstream analysis showed that stringent read count filtering leads to high-quality data. We show that stripe artifacts, which are visible on the raw count data, can be effectively removed using standard normalization methods[51]. However, the high stringency led in some ST images to the removal of whole column or row elements. Both, the so-called stripe artifacts and the empty rows or columns, result most likely from heterogeneous flow conditions within the horizontal or vertical channels of the PDMS chip. This could be further optimized by introducing standard fluid interfaces to the chip, ensuring homogenous fluid flow, and handling of air bubbles through micromechanical features in addition to the already implemented bubble traps at the transition from inlet to channel.

The presented results show high-quality spatially resolved transcriptomic datasets from kidney, cerebellum, heart, spleen, and kidney. With this we demonstrated that xDBiT is suitable for a variety of tissues, which will facilitate studies focused on complex diseases and multi-organ dysfunction. Only for the pancreas sections xDBiT showed low read and gene counts. This can be explained by the high RNAse content of the pancreatic acinar cells and suggests that the development of an optimized mRNA preservation protocol would be required to investigate pancreatic tissue[52].

Despite the ST technology advances reported here and by others, the lateral diffusion of molecules in the barcoding step of ST methods, limits the resolution of barcoding-based ST methods to the range of 5–10 μm[53]. However, ST datasets with subcellular resolution require elaborate algorithms to segment single cells based on the spatial transcriptome[6]. Rather than further increasing the resolution of spatial transcriptomic methods, an alternative approach is the use of single-cell transcriptomic datasets and computational methodologies to increase the resolution of the datasets in silico. Thus, a large and complex experimental design with the objective of mapping cell transcriptomes and retaining tissue context requires more affordable technologies. Here, we have shown that xDBiT is a low-cost ST technology (ca. 125 € per sample) that provides robust and accurate analysis of spatial gene expression patterns. The achieved transcript read depth on xDBiT spots, together with deconvolution tools, is sufficient to resolve rare cell types, such as podocytes in the glomeruli of the mural kidney. Thus, xDBiT is an ST methodology that optimally balances the cost and throughput. Further engineering efforts will focus on increasing the screening areas, in addition to read depth. The xDBiT workflow could be further scaled to larger screening areas by increasing the microfluidic channel length as well as the microfluidic chip platform. One limiting factor of the xDBiT approach is the fluid resistance, which scales linearly with the channel length. From our correlation analysis between the fluid flow rates and microchannel lengths in the xDBiT PDMS chips, we can conclude that microchannels with <260 mm length can be operated under the chosen pressure conditions. Thus, we anticipate that a higher degree of multiplexing than presented here could be achieved. Longer channels would require, however, a higher fluid forward pressure to drive fluid flow, which in turn would induce leakage between the microchannels and disruption of the underlying tissue.

Furthermore, barcoding strategies with microfluidic channels can be combined with a multitude of modalities, including DNA-barcoded antibodies[17], chromatin accessibility[19], and epigenomic readouts[18]. To increase adaptability, xDBiT libraries can be sequenced using standard next generation sequencing platforms. Deterministic barcoding can also be performed with archived formalin-fixed and paraffin-embedded (FFPE) samples, however with lower read depth[54]. This is expected due to the fact that FFPE-derived RNA is highly degraded and chemically modified, and affects downstream sequencing processes[55,56].

Since the microfluidic workflow has adverse effects on the integrity of the tissue sections and image information is needed to further enhance the power of spatial transcriptomic data[25], we introduced two imaging steps to allow the acquisition of high-quality image data. This allows the platform being used for the analysis of high-resolution image features in conjunction with transcriptomic information.

Finally, in addition to technical advances, we have provided an open-source analysis pipeline to generate xDBiT datasets and make the method easily accessible. This includes a semi-automatic image registration pipeline and the introduction of alignment marks to robustly align the fluorescent images with ST data. In summary, using xDBiT, we expanded the toolbox of spatial transcriptomic methods for higher throughput measurements and improved both the transcriptomic and image quality of the resulting datasets.

## Methods

### Ethics statement

Animal experiments were carried out in compliance with the German Animal Protection Act and with the approved guidelines of the Society of Laboratory Animals (GV-SOLAS). All animal used within this study were kept at the HMGU Core Facility Laboratory Animal Services (CF-LAS), Neuherberg, Germany. All procedures were carried out in compliance with German Animal Welfare Legislation and the regulations of the Government of Upper Bavaria, Germany. Animal housing was approved according to §11 of the German Animal Welfare Act and performed in accordance with Directive 2010/63/EU.

### Husbandry and tissue collection

Wild-type C57BL/6 J mice were purchased from Charles River UK Ltd (Margate, United Kingdom) and were maintained under specific pathogen-free conditions under strict 12 h dark-light cycles. All mice were kept in a positive pressure system, maintaining a temperature between 19 and 23 °C, 55% humidity, and had free access to water and standard mouse chow diet.

Three male C57BL/6 J mice (age 3–4 months) were used in the multi-organ study. For the cross-contamination experiment, two male C57BL/6 J mice (ages 3 months and 22.5 months) were used. At the time of experiment, mice were sacrificed in accordance to GV-SOLAS regulation, and were subsequently dissected. Heart, liver, kidney and spleen were collected from the same two mice while the brain sample was collected from a different mouse. The organs cryo preserved using Tissue-Tek OCT Compound (CellPath Ltd, UK) into Tissue-Tek Cryomolds (Sakura Finetek, USA). All cryo embeddings were frozen in pre-chilled 2-methylbutane on dry ice. After freezing, cryo embeddings were transferred into −80 °C freezer for long term storage. For the brain, cerebrum and cerebellum were embedded separately.

## Statistics and reproducibility

In total, two independent xDBiT experiments with each nine tissue sections from six organs (heart, kidney, liver, spleen, pancreas and cerebellum) have been performed. A detailed information on the samples can be found in Supp. Table 2. No statistical method was used to predetermine the sample size. The pancreas samples were excluded from downstream analysis due to low read counts. The experiments were not randomized and the investigators were not blinded to allocation during experiments and outcome assessment.

## Master mold fabrication

Master molds for the horizontal and vertical PDMS chips were fabricated according to standard SU-8 (SU-8 3050; Microresist Technology, Germany) photolithography protocols[57]. To prevent PDMS adhesion to the SU-8 mold, the surface was spin-coated with a thin film (<1 μm) of CYTOP™ (AGC Chemicals, Japan). To evaporate the CYTOP™ solvent the SU-8 mold was heated to 160 °C for 1 h.

## Horizontal and vertical microfluidic chip fabrication

Horizontal and vertical microfluidic PDMS chips were manufactured by casting a 5 mm PDMS (Sylgard® 184, Dow Corning, MI, USA) layer (ratio 10:1 of base material to curing agent) onto the SU-8 master mold. After degassing for 1 hour in an evacuated desiccator at room temperature (rt), the PDMS was cured for 1 h at 80 °C. The cured PDMS chip was peeled off, cut into the required size and inlets and outlets were punched using a 14 gauge needle.

## Fabrication of non-PDMS adapters

To press the PDMS chips onto the tissue sections, a plastic clamp was milled in acrylic glass. Well adapters that allow the precise application of reagents onto the tissue sections and molds for PDMS gaskets were 3D-printed with a DLP stereolithography printer (Pico2HD, Asiga, Australia) using the resin PlasGRAY (Asiga, Australia). Printing parameters including light intensity and exposure time were set according to the manufacturer's material file. After exposure, the printed part was removed and sonicated in isopropanol for 10 min. Afterwards, the printed parts were incubated for 4 h at room temperature to remove excess isopropanol and post-cured at 2000 flashes per side (Otoflash curing unit).

## Fabrication of PDMS adapters and gaskets

PDMS gaskets and the vacuum adapter were manufactured by replica molding using 3D printed molds. After 3D printing as described above, the molds were dip-coated with CYTOP™ (AGC Chemicals) and incubated on a hotplate for 8 h at 80 °C. For gaskets, a 5 mm layer of PDMS (ratio 10:1 of base material to curing agent) was poured into one well of a 6-well plate. The mold was pressed upside down into the PDMS and the material was degassed for 1 h in an evacuated desiccator at rt. After curing for 1 h at 80 °C, the PDMS gasket was peeled off carefully and excess material was removed with a knife. For the vacuum adapter, the mold was glued to the bottom of a well and PDMS was poured over the mold. Degassing and curing was performed as described and a hole was punched in one of the sides using a 2 mm punching needle. An overview of all modules required in the xDBiT workflow is shown in Supp. Fig. 1 A–G.

## Flow rate measurement

To measure the flow rate in the different channels of the xDBiT PDMS chip, a horizontal PDMS chip was mounted on a glass object slide. 10 μL of food color dyed water was added to all inlets. Vacuum was applied to the outlets until all channels were filled completely. Then, the outlets were emptied using a vacuum aspirator. To start the measurement, the vacuum adapter was attached to the outlets and 300 mbar vacuum were applied for 60 seconds. Then, the vacuum adapter was removed and the volume in the outlet channels was measured using a 10 μL

pipette. This procedure was conducted 2 times, yielding four data points per channel length. Linear regression was performed using the Python package *statsmodels* (v0.12.2) with the ordinary least squares method.

## Tissue preparation

For sectioning, the organs were warmed to −15–18 °C inside the cryostat (Leica). Object slides with marked placement areas were cooled inside the cryostat before use for at least 5 minutes. The tissue blocks were sectioned with a thickness of 10 μm using RNAse free equipment, placed in predetermined positions on the object slide (Supp. Fig. 1H) and attached by warming the backside of the object slide with a finger. The sectioned samples were stored at −80 °C.

## Generation of optimal RevT primer barcode sets

To prevent reverse transcription bias from RevT primer barcodes, we used mixes of multiple RevT primers in the RevT reaction. BARCOSEL[58] was used to generated nine sets of RevT primers with 4 barcodes per set (Supplementary Data 2). For the cross-contamination analysis experiment these nine sets were used individually for each sample well. In the multi-organ experiments we did not barcode the wells in the RevT step separately and instead mixed sets 1-4 to further increase the diversity. The RevT primers (Supplementary Data 1, Sigma) were dissolved in ultrapure water at a concentration of 100 μM and mixed at this concentration.

## Preparation of ligation barcoding plates

A complete list of the barcoded ligation oligos used for xDBiT can be found in Supplementary Data 3 and 4. The ligation oligos were dissolved in ultrapure water at a concentration of 100 μM and stock plates were stored at −20 °C. Separately for ligation round #1 and ligation round #2, 36 ligation barcode oligos were annealed with the respective bridge oligo (Supplementary Data 5). In brief, 21.1 μL of a bridge oligo (1 mM, Sigma) were mixed with 296 μL water and 317 μL 2× annealing buffer (5 mM Tris, 100 mM NaCl) to a final concentration of 33.33 μM. Then, 4 μL of one ligation barcode oligo (100 μM) and 12 μL of the diluted linker were mixed in a 96 well plate. Using a PCR cycler, the oligos were denatured at 95 °C for 2 minutes and cooled to 20 °C at a rate of −0.1 °C/s to anneal the strands. The annealed oligo stock plates were stored at 4 °C for short-term or −20 °C for long-term storage. Before the experiment, 1 μL of each barcode was distributed to fresh PCR plates, later called 'Ligation Barcoding Plate' #1/#2.

## Fixation, permeabilization and blocking

The object slide with tissue sections was thawed at 37 °C for 1 min on a heated plate. Clamp, 1-well adapter and PDMS gasket (Supp. Fig. 1E) were assembled, aligned and attached to the tissue slide. The tissue sections were washed with 1× RNAse-free phosphate buffered saline (PBS, Invitrogen) supplemented with Murine RNAse inhibitor (1 U/μL, "RI", New England Biolabs) and ribonucleoside vanadyl complex (RVC, 10 mM, New England Biolabs) and fixed in 4% paraformaldehyde (PFA, Sigma) for 40 min at room temperature (RT). After three washes in 1× PBS complemented with RVC (10 mM, "PBS + RVC"), the tissue sections were permeabilized with 0.2% Triton X-100 (Sigma) in PBS + RI for 10 minutes at RT and blocked for 30 min at RT with 1% bovine serum albumin (BSA, Thermo Fisher).

## Staining and high-resolution confocal imaging

The CD31 primary antibody (Thermo Fisher, PA5-16301) was diluted 1:50 in antibody diluent (PBS + RI supplemented with 0.1% Tween-20 and 3% donkey serum), added to the sections and incubated for 30 min at RT. After 3X wash in PBS-T (0.1% Tween-20) supplemented with RVC (PBS-T + RVC), nuclei, actin filaments and primary antibody were stained using DAPI (1.25 μg/mL, Sigma), Phalloidin-iFluor647 (1.25×, Abcam) and AF555 secondary antibody (Invitrogen, A-31572; dilution:

1:500) in antibody diluent for 30 minutes at room temperature in the dark. The tissue sections were washed three times in PBS-T supplemented with RI (PBS-T + RI) and mounted in 85% ultrapure glycerol (Sigma) supplemented with 2 U/μL RI using #1.5 coverslips (Menzel). Images were acquired using an LSM 880 confocal microscope (Zeiss) with a 20×/0.8 objective (Zeiss) at a final resolution of 0.24 μm/pixel using the ZEN 2.3 SP1 FP3 (black) software.

### Reverse transcription (RevT)

The coverslip was removed by holding the object slide in a 45° angle with the coverslip facing down into 3X saline sodium citrate (SSC) buffer until the coverslip falls off. The sections were dipped 3X in ultrapure water and dried under airflow. Clamp and 9-well adapter (Supp. Figure 1F) were assembled, aligned and attached to the tissue slide. PBS + RI supplemented with 1% BSA was added and stored at 4 °C for maximum 30 minutes until the next steps were performed. An RevT reaction mix was prepared from 514.8 μL ultrapure water (Thermo Fisher), 158.4 μL RevT buffer (5×, Maxima H Minus RT Kit, Thermo Fisher), 39.6 μL dNTPs (10 mM, New England Biolabs), 19.8 μL RI, 19.8 μL RevT primer set and 39.6 μL Maxima H Minus Reverse Transcriptase (200 U/μL, Thermo Fisher). A total of 80 μL of the mix were added to each well, the wells were sealed and the slide was incubated for 30 minutes at RT and 90 minutes at 42 °C in a closed thermoshaker without agitation. To ensure equal heat distribution and minimize evaporation an aluminum block was placed between object slide and hot plate and wet tissues were added to the closed container. Afterwards, the tissue sections were washed once in PBS-T + RVC and the 9-well adapter was removed.

### Spatial barcoding by ligation (horizontally or vertically)

The object slide was dipped 3× into ultrapure water to remove salts and the tissue sections were dehydrated stepwise by incubation in 70, 85, and 99.5% ethanol for 1 min each and dried briefly under airflow. The horizontal (ligation round #1) or vertical (ligation round #2) PDMS chip was aligned, attached to the tissue sections and placed into the clamp (Supp. Figure 1D) and the screws were tightened uniformly and strongly to prevent leakage.

To rehydrate the tissue, 5 μL of PBS + RI were added to each inlet and the channels were filled by applying 300 mbar vacuum to the outlets using a PDMS vacuum adapter (Supp. Fig. 1G) and incubated for about 10 min at RT. A ligation reaction master mix was prepared from 149.66 μL ultrapure water, 26.3 μL T4 DNA Ligase buffer (New England Biolabs), 2.51 μL 10% Triton X-100 (Sigma), 13.1 μL Murine RNAse inhibitor, 5.25 μL Tartrazine (10 mg/mL, Carl Roth) and 13.2 μL T4 DNA ligase (New England Biolabs). 4 μL of the master mix were added to the Ligation Barcoding Plate #1 or #2 (see above) respectively for a total of 5 μL and centrifuged down briefly.

The inlets of the PDMS chip were emptied using a vacuum aspirator with attached pipette tip and 5 μL of each barcode was added to the inlets according to the inlet filling scheme (Supp. Table 1). The outermost channels were filled with an alignment marker mix consisting of 80 μg/mL anti-BSA antibody (Invitrogen) in antibody diluent. To remove air bubbles in the inlets, the chip was centrifuged at 100 × g for 1 min. The channels were filled using vacuum as described before. Inlets and outlets were sealed and the chip was incubated at 37 °C in a closed thermoshaker without agitation. To ensure equal heat distribution and minimize evaporation an aluminum block was placed between object slide and hot plate and wet tissues were added to the closed container. After 15 min the vacuum was applied again to remove air bubbles in the channels and the chip was incubated another 15 min at 37 °C for a total of 30 min reaction time. The inlets were emptied with the vacuum aspirator and the channels were washed for 5 min with PBS-T + RI. Afterwards, the channels were emptied and the chip was removed.

### Secondary staining and alignment imaging

The alignment markers were stained with 4 μg/mL donkey anti-rabbit AlexaFluor 555 secondary antibody (Invitrogen, A-31572) in PBS-T + RI supplemented with 3% donkey serum, 1.25 μg/mL DAPI and 1.25× Phalloidin-iFluor647 for 30 minutes at room temperature in the dark. Afterwards, the tissue sections were washed three times in PBS-T + RI and mounted as described before. Images were acquired using an LSM 880 confocal microscope (Zeiss) and a 20×/0.8 objective (Zeiss) at a final resolution of 0.49 μm/pixel using the fastest possible scanning mode.

### Lysis and sample collection

The coverslip was removed from the tissues and the 9-well adapter attached as described before. Lysis buffer was prepared from 10 mM Tris-Cl pH 8.0, 200 mM NaCl (Sigma), 50 mM EDTA pH 8.0 (Life Technologies), 2% SDS (Bio-Rad) and 2 mg/mL proteinase K (New England Biolabs). The tissue sections were lysed separately in 75 μL lysis buffer for 2 h at 55 °C. To prevent evaporation, the wells were closed with a PDMS piece which was fixed with tape and incubation was conducted in a closed container containing wet tissues. Afterwards, possibly remaining parts of the tissue sections were scraped off with the pipette tip and the lysates were collected in nine separate DNA LoBind tubes (Eppendorf). The wells were washed once with 40 μL of lysis buffer and the washing solution was pooled with the lysate. Samples were stored at −80 °C.

### cDNA purification

396 μL of Dynabeads MyOne Streptavidin C1 (44 μL per sample, Thermo Fisher) were washed three times in 800 μL 1× B&W buffer (see manufacturer's manual) supplemented with 0.5% Tween-20 and 0.05 U/μL RI and resuspended in 950 μL of 2× B&W buffer supplemented with RI (100 μL + 5% per sample). The lysates were thawed at rt, brought to 100 μL with ultrapure water and 5 μL PMSF (200 mM, Cell Signaling) were added and incubated for 10 minutes at rt to block Proteinase K activity. To bind the cDNA to the beads, 100 μL of the resuspended Dynabeads were added to the lysates, vortexed and incubated for 1 h at rt under agitation (1200 rpm). Afterwards, the beads were washed two times in 1× B&W-T + RI for 5 min at rt under agitation. Likewise, a final washing step was performed in 10 mM Tris-Cl pH 8.0 buffer supplemented with 0.01% Tween-20.

### Template switch

A template switching reaction mix (TSR mix) was prepared from 360 μL ultrapure water, 180 μL RevT buffer (5×), 180 μL Ficoll PM-400 (20%, Sigma), 90 μL dNTPs (10 mM), 22.5 μL Murine RNAse inhibitor, 22.5 Template Switching Oligo (Supplementary Data 5, 100 μM, Ella Bioscience) and 45 μL Maxima H Minus Reverse Transcriptase (200 U/μL). The beads with the bound cDNA were placed against a magnetic rack and washed once in ultrapure water. Then, the beads were resuspended in the TSR mix and incubated for 30 min at RT and 90 min at 42 °C under agitation (1200 rpm). Afterwards, the samples were placed against a magnetic rack and washed once in ultrapure water.

### PCR amplification

A PCR mix was prepared from 869 μL ultrapure water, 1034.6 μL Kapa Hifi 2X Master Mix (Roche), 82.8 μL cDNA amplification forward primer (10 μM, oSR321211_TSO_fwd, Supplementary Data 5) and 82.8 μL reverse primer (10 μM, oSR321212_TSO_rev, Supplementary Data 5). Each sample was resuspended in 220 μL PCR mix and split equally into 4 different PCR tubes. PCR was performed using following program: 95 °C for 3 min, then 5 cycles of 98 °C for 20 s, 65 °C for 45 s and 72 °C for 3 min. Afterwards, the reaction mixtures were pooled and placed against a magnetic rack. 200 μL of each sample were transferred to a fresh tube and 2 μL of SYBR Green qPCR dye (100 μM, Jena-Bioscience)

were added. To account for differences in the cDNA content between the samples an optimal number of PCR cycles was determined for each sample separately. Duplicates of 10 μL of each sample were transferred into a qPCR plate and measured in a Viia 7 qPCR machine (Applied Biosystems) using following program: 95 °C for 3 min, then 40 cycles of 98 °C for 20 s, 67 °C for 20 s, 72 °C for 1 min. The optimal cycle number was defined as the cycle where the qPCR curve reaches 25% of its maximum intensity. The remaining 180 μL per sample were distributed into two PCR tubes and the following qPCR program was run with the previously determined cycle number n: 95 °C for 3 min, then n cycles of 98 °C for 20 s, 67 °C for 20 s, 72 °C for 3 min, and a final extension at 72 °C for 5 min, then hold at 4 °C. Afterwards, the qPCR reactions were pooled per sample.

## cDNA purification
The amplified cDNA was purified using SPRIselect™ beads (Beckman Coulter) following a left sided size selection with a bead-to-sample ratio of 0.8×. In brief, 160 μL of sample were mixed with 128 μL of resuspended SPRIselect beads and incubated for 5 min at rt. Beads were washed two times in 85% ethanol and air-dried for 3 min. The cDNA was eluted in 20 μL ultrapure water by incubation at 37 °C for 10 min. The supernatants were transferred to a fresh tube resulting in 9 tubes of purified cDNA. The quality of the cDNA was analyzed using the Bioanalyzer High Sensitivity DNA chip (Agilent) and samples were stored at −20 °C.

## Library preparation and sequencing
The concentration of the cDNA was determined using a Qubit 1× dsDNA assay (Invitrogen) and the sequencing library was generated using the Nextera XT DNA Library Preparation Kit (Illumina). The quality of the library was assessed using the Bioanalyzer High Sensitivity DNA chip (Agilent). Samples were sequenced on a NovaSeq 6000 system (Illumina) at a sequencing depth of minimum 50,000 reads per spot using a 100 cycles kit in paired-end mode. Following read length configurations were used: R1: 74 cycles, i7: 6 cycles, R2: 58 cycles. Importantly, when using the discussed double barcoding approach, a 200 cycles kit is required. The settings are then, R1: 100 cycle, i7: 6 cycles, R2: 100 cycles.

## Pipeline overview
Integration of sequencing results and imaging data was performed using a custom pipeline which is published open-source on Github (https://github.com/jwrth/xDBiT_toolbox) and combines two previously published analysis pipelines: *Drop-seq tools* v2.1.0[59] and *splitseq_toolbox*[60] with custom *Python* and *Bash* script. Further, it uses functions from the *Picard* toolbox[61]. The pipeline consists of 2 main steps: (1) *ReadsToCounts* and (2) *CountsToAnndata* (Fig. 1D). The first part of the pipeline needs to be run on a Linux machine while the second part was tested both on a Linux and Windows machine. In the following sections the pipeline is explained briefly. Detailed instructions to process xDBiT data can be found in the Github repository. For plotting the Python packages *matplotlib* v3.5.1[62] and *seaborn* v0.11.2[63] were used. Image transformations were predominantly performed using the *OpenCV* package[64].

## ReadsToCounts
This script takes two FASTQ files (Read 1 and Read 2) and barcode-coordinate information as input and processes them as follows: Read 1 sequences are trimmed and filtered using cutadapt v3.7[65] and mapped against the mm10 (GRCm38) mouse genome using STAR-2.7.4a[66]. Unique molecular identifiers (UMIs) and spatial barcodes are extracted from Read 2 using the Drop-seq tool *TagBamWithReadSequenceExtended*. A custom Python 3 pipeline, using *samtools* (v1.9)[67] and *pysam* (v0.19.1)[68], assigns coordinates using barcode information provided in a CSV file. Reads without a valid x- or

y-barcode are discarded in this step. The *DigitalExpression* function is used to collapse the UMIs and generate a spot/gene count matrix. RNA metrics are calculated using *CollectRnaSeqMetrics*. Importantly, by running the *ReadsToCounts* pipeline in 'xDbit' mode, it also takes the RevT barcode (z-barcode) into account. This allows the removal of potential cross-contaminations.

## CountsToAnndata
In this step the spot/gene count matrix and imaging data are aligned and integrated. In brief, the positions of the alignment marker vertices are extracted semi-automatically from the alignment images using *napari*[69] and *Squidpy* (v1.1.2)[23]. The coordinates of the vertices are used to register alignment image and xDBiT spots by performing an affine transformation using *OpenCV*[64]. In order to align the high-resolution images of the first imaging round with the xDBiT spots, the SIFT algorithm[22] is used to extract common features between the alignment DAPI image and the high-resolution DAPI image. Based on the coordinates of these features an affine transformation matrix is determined, which is used to align the xDBiT spots to the high-resolution image. The dataset is saved in the *AnnData* format[20]. In this study we included intronic reads (Supp. Fig. 5D) into the analysis.

## Sequencing saturation analysis
To investigate the saturation of the sequencing runs, we subsampled the sequencing reads before library construction using the subsampling feature of *samtools view* (v1.9)[67]. To run the analysis on multiple files using multiple cores, python and Bash scripts were developed, which can be found on https://github.com/jwrth/xDBiT_toolbox/ReadsToCounts/subsampling (v2.1) together with a more detailed instruction on the commands to be used. The analysis has been tested on a Linux system.

## Preprocessing
Pre-processing of the count matrices was performed using the Python 3 tools Scanpy v1.8.2[21] and Squidpy v1.1.2[23]. To remove the background, we excluded spots with a mean DAPI signal below a certain threshold. Removed background spots were used to filter out all genes that had a mean background expression $\mu_b$ below a threshold $t_g$ in all samples. The threshold $t_g$ was defined as:

$$t_g = \mu_b + 2 \cdot SD_b$$

$$\text{with } SD_b = \sqrt{\mu_b}$$

We assumed a Poisson distribution of the background read counts and calculated an approximation of its standard deviation SD. To estimate the remaining cross-organ spillover we selected 100 specific genes per organ from the HOMER database[27]. The xDBiT datasets were grouped by organ and we calculated per organ how many of the 100 organ-specific genes are present in the individual xDBiT datasets. All further analyses were performed according to current best practices in single-cell RNA-seq and Spatial Transcriptomics analysis[23,51] and can be reproduced using Jupyter Notebook (for more information see Data and Code availability). Counts were normalized, log-transformed, and the top 2000 highly variable genes were determined. Batch correction was performed per section using *scanorama*[70] (v1.7.2).

## Dimensionality reduction and clustering
For visualization in lower dimensional space, we calculated the top 50 principal components and generated a two-dimensional representation using Uniform Manifold Approximation and Projection (UMAP)[28]. To group the spots into transcriptomically similar clusters the Leiden algorithm[29] was applied. Overlay plots of transcriptomic spots and image data were generated using a custom plotting function.

## Differential gene expression analysis

Differentially expressed genes for each Leiden cluster were calculated by applying Scanpy's *rank_genes_groups* using the Wilcoxon rank-sum test and default settings. The top 3 differentially expressed genes were visualized using *rank_genes_groups_heatmap*. For downstream analyses the 300 most significantly differentially expressed genes were used. Information about protein expression of differentially expressed genes in the respective tissues has been taken from The Human Protein Atlas[71].

## Gene Ontology (GO) term enrichment analysis

For GO term enrichment analysis we used APIs of the STRING web server[34]. A detailed description on the how the enrichment is calculated can be found in ref. [72]. The resulting False Discovery Rate (FDR) shows p-values corrected for multiple testing using the Benjamini-Hochberg procedure. Enrichment scores are represented as negative $\log_{10}$ of the FDR. For our analysis we searched for enrichments in the *Brenda Tissue Ontology* database (BTO)[35] and the *Biological Processes* GO database[73,74].

## Cross-contamination testing in 9-well adapter

The occurrence of potential cross-contaminations between the wells of the 9-well adapter was tested using water colored with "golden yellow" and "royal blue" icing color (Wilton). The icing color was added to the water until it reached the desired color. The 9-well adapter was attached to an empty object slide as sown in Supp. Fig. 1F) and the colored water was added in a checkered pattern. Photos were taken using a Canon PowerShot SX620 HS digital camera before and after 24 h incubation at room temperature.

## Analysis of post-RevT cross-contaminations

To test for cross-contaminations between the samples occurring after the RevT, an experiment with eight liver sections was performed, leaving the center well free. Samples were barcoded twice, i.e., during the RevT reaction at the beginning of the xDBiT workflow (Fig. 1A III), and during library preparation of the individually retrieved samples at the end of the xDBiT workflow (Fig. 1A VI + VII). Unique barcodes were added via RevT primers (see Supplementary Data 2) and indexing primers, respectively. The xDBiT experiment was performed using the standard protocol described above and a library preparation for all nine wells was performed. From the center well we were not able to retrieve enough cDNA for library preparation and sequencing. Sequencing was performed on a NovaSeq 6000 system (Illumina) using a 200 cycles kit. The used settings were R1: 100 cycle, i7: 6 cycles, R2: 100 cycles. To calculate the percentage of cross-contamination reads, only read 2, containing the RevT barcodes and spatial barcodes were analyzed. For the analysis the *ReadsToCounts* script was modified to disregard read 1 and instead only run up to the barcode filtering steps to retrieve counts of the spatial barcodes and RevT primer barcodes. This script can be invoked using the '–spatial_only' flag. Further, to be able to catch information about reads from other wells, information about all 36 barcodes used in the experiment was added to the barcode legend file. Count values of the found barcodes were stored in the 'recording_dictionary.json' file in the 'rna_out' folder. From this information, the percentage of RevT barcodes in the different wells were calculated. The Jupyter notebook showing the analysis is provided in the Github repository.

## Correlation with bulk sequencing data

To compare xDBiT ST data with published bulk sequencing data we generated a pseudobulk dataset of the xDBiT dataset by summing up the counts of all spots per gene. Bulk RNA-sequencing datasets were downloaded from the ENCODE project website[36,37] and are listed in Supp. Table 3. Both the bulk and the pseudobulk datasets were normalized to transcripts per million (TPMs) and log transformed. To analyze the correlation of datasets per organ the Pearson correlation coefficient was calculated pairwise and results were visualized as heatmap.

## Comparison with published DBiT-seq datasets

Previously published DBiT-seq datasets from embryonic sections[16] were downloaded from the Gene Expression Omnibus database with the accession code GSE137986. Of the whole dataset following experiments were retrieved for the comparison: GSM4189613 (Embryo stage E10−162,684,631 raw reads) and GSM4189612 (Embryo stage E12−53,619,846 raw reads). In addition to the xDBiT datasets, we used for comparison (1) a dataset that was generated in-house following the protocol of the original DBiT-seq method and (2) a dataset that was generated using the original DBiT-seq PDMS chip without serpentine channels but with the optimized biochemical protocol of xDBiT. All datasets were normalized to the total number of raw sequencing reads and then compared by the normalized values of total counts per spot and number of genes per spot.

## Image processing

For image processing and generation of figures, we used Fiji ImageJ v1.53c[75] and the Quickfigures toolkit[76]. Stitching of the tiled images was performed using a custom ImageJ script utilizing the Grid/Collection Stitching algorithm[77].

## MEFISTO factor analysis

MEFISTO factor analysis[38] was performed using the Python package *mofapy2* (v0.6.4). Datasets of each tissue section were analyzed separately. Spatial spot coordinates were used as covariates and only highly variable genes were selected for the analysis. Following parameters were used for the analysis: factors=10; frac_inducing: 0.5; sparseGP=True; start_opt=10; opt_freq=10. Models were saved as hdf5 files and downstream analysis was performed using the *mofax* toolbox (https://github.com/bioFAM/mofax). To investigate the first four factors functionally, for each factor the top weighted genes (> 95 confidence interval) were selected and used for GO term enrichment analysis using the STRING algorithm[34] as explained above.

## Cell type mapping in xDBiT kidney data

To map the cell types from single-cell datasets onto xDBiT spatial transcriptomics data of the murine kidney, we applied *cell2location* (v0.1)[13]. The single-cell RNA-seq dataset was retrieved from a previous publication including P0 and adult mice samples[41]. For the analysis, only cells from adult mice were selected and mitochondrial genes were removed from both the single-cell and the representative xDBiT kidney dataset. Genes were filtered using the cell2location gene_filter function, filtering out genes that were detected in less than five cells and less than 0.05 % of cells. Anndatas were prepared for analysis using *scvi-tools* (v0.16.4)[78]. The single-cell model to infer expression signatures of cell types was trained in 250 epochs. Spatial mapping was performed with default parameters, except for: N_cells_per_location=20; detection_alpha=20; max_epochs=30000; batch_size=None; and train_size=1. To show the minimum number of cells, we used the 5% quantile of the resulting posterior distribution, reflecting the confidently predicted number of cells.

## Reporting summary

Further information on research design is available in the Nature Portfolio Reporting Summary linked to this article.

# Data availability

All raw sequencing data and preprocessed xDBiT data, including spatial transcriptomic data with aligned images as well as data of the cross-contamination studies, have been deposited and are publicly available on GEO under the accession number GSE207843. CAD plans to manufacture the xDBiT master molds using photolithography as well as the

plans of the 3D-printed and milled parts necessary for the workflow are stored in the 'cad' folder of the publicly available Github repository https://github.com/jwrth/xDBiT_toolbox (v2.1)[79]. All source data, that is not sequencing data and necessary to replicate the figures, is deposited in the Github repository under 'publication/source_data'. Data for GO term enrichment analysis was obtained from *Brenda Tissue Ontology* database (BTO)[35] and the *Biological Processes* GO database[73,74]. Protein expression data was obtained from The Human Protein Atlas[71]. Previously published DBiT-seq datasets were obtained from GEO under following accession numbers: GSM4189613 (Embryo stage E10 – 162,684,631 raw reads) and GSM4189612 (Embryo stage E12 – 53,619,846 raw reads). Bulk polyA plus RNA-seq data was obtained from ENCODE database: ENCSR000CGZ (Heart), ENCSR000CHA (Kidney), ENCSR000CGW (Spleen), ENCSR966JPL (Spleen), ENCSR000CHB (Liver), ENCSR000CGX (Cerebellum). The kidney single-cell RNA-seq dataset used for deconvolution was retrieved from GEO under the accession number GSE157079.

## Code availability

All code, including notebooks, functions and environment files with package versions to rerun the analysis, is publicly available in the Github repository https://github.com/jwrth/xDBiT_toolbox (v2.1)[79]. The computational pipeline, consisting of the scripts *ReadsToCounts* and *CountsToAnndata* can be found in the subfolders with the corresponding names. ImageJ scripts to stitch images from tile scans are deposited in the folder named 'imagej'.

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

## Acknowledgements

We thank T. Gerlach and J. Promoli for milling the acrylic glass parts at the workshop of Helmholtz Munich. We thank Inti I. A. de la Rosa Velazquez and G. Eckstein for performing the NovaSeq sequencing at the Bioinformatics Core Facility of Helmholtz Munich and T. Walzthöni for the bioinformatics support. We thank S. Kublik for performing the NextSeq sequencing in the early project stages at the Research Unit Comparative Microbiome Analysis of Helmholtz Munich. This work was supported by the Helmholtz Pioneer Campus and ERC Consolidator Grant (Number 772646). Figures and schematics were created using Affinity Designer 2. Third-party icons were retrieved from flaticon.com.

## Author contributions

J.W., N.H., C.P.M.-J., and M.M. designed the study. J.W. and N.H. designed the xDBiT microfluidic chip platform and the 3D-printed modules. N.H. and S.B. manufactured the microfluidic PDMS chip and the 3D-printed adapters. J.W. and S.B. developed the experimental method of xDBiT. J.W. developed the imaging strategy and the computational pipeline. K.Y. and S.C. collected the murine organ samples. J.W. performed all xDBiT experiments, processed the raw data and performed all downstream computational analyses. M.M. and C.P.M-J. received the funding and supervised the study. The manuscript was written by J.W., M.M. and C.P.M-J. All authors corrected and approved the manuscript.

## Funding

## Competing interests

The authors declare no competing interests.
