## [Peer Review File · Nature Communications]

Spatial Transcriptomics Using Multiplexed Deterministic Barcoding in TissueREVIEWER COMMENTS

Reviewer #1 (Remarks to the Author):

Wirth et al present multiplexed deterministic barcoding in tissue (xDbit), an extension of previous technology DBiT-seq, which profiles spatially resolved transcriptomes of up to nine distinct tissues in a single experiment. The authors improve on DBiT-seq by enabling multiplexing of up to nine tissues via laying out horizontal and vertical axis channels in a serpentine fashion, alongside a 3D printed 9-well chip to separate out the tissues for sample labelling prior to next generation sequencing. Enabling transcriptome-scale spatial technologies at much lower cost per sample is key to widespread use of spatial omics in hypothesis-driven and clinical research. The authors demonstrate the capability of xDbit by profiling a series of adult mouse tissues including brain, liver and spleen and compare sensitivity to the previous DBiT-seq approach. The authors also perform state-of-the-art techniques including MEFISTO and cell2location to demonstrate the ability of extracting insights and integrating with existing dissociated transcriptomics data respectively with scripts and processing software made available via Github. The manuscript is well motivated and written extremely clearly. I have some additional comments and questions that I think should be addressed.

- Is there any risk of spillover of signal among the 9 samples? This can be checked by correlation of expression profiles of the vertical and horizontal barcodes across the 9 samples. Providing some diagnostic plots along these lines would be very insightful. It may be desirable to design a benchmarking experiment where one can assess the proportion of incorrectly assigned reads to samples or tissue areas.
- Are there any transcript sequences without and x-, y-, or either x-y barcodes in the resulting datasets? i.e. tissues not coming from the PDMS channels? If these transcripts are retained in the final dataset are they used for any further analysis or discarded? Describing this further may clear any confusion for readers.
- The abstract states 18 spatially resolved transcriptomics datasets were captured, which suggests to me that 2 9-tissue xDbit experiments were performed. The authors should assess any systematic biases detected among these. Figure 3F does display the concordance of expression between the samples, however it's not immediately clear which experiment (same or different) the samples came from.
- In Figure 4c we observe some horizontal and vertical stripe artefacts. This suggests that some of the channels in the PDMS chip may not work all the time. I would suggest discussing this potential limitation as well as discussing potential ways to address this, either experimentally or algorithmically.
- I suggest the authors perform the MEFISTO analysis presented in Figure 4 separately per tissue. It's unclear to me why Factor 1 would be shared with brain, spleen and to some extent kidney. It may be more insightful examining the factors determined from each tissue.
- How applicable is xDbit to archival tissues, e.g. those preserved using FFPE technique?
- How applicable is xDbit to histopathological assessment before or after xDbit workflow? e.g. H&E staining and imaging prior to the immunofluorescence staining described.
- line 426 "Error! Reference source not found."

Reviewer #2 (Remarks to the Author):

Wirth et al present a multiplexed approach, xDbit, of the previously published Dbit seq method.

The manuscript is well written and easy to follow. The authors also perform the necessary comparisons to the Dbit seq technology. The authors also present some novel tools for data analysis. The methodology is sound.

The main benefit with the new approach is the ability to multiplex several tissue sections in one experiment. That is an important feature and would facilitate a higher throughput of experiments using Dbit seq methodology.

Since this is the main finding in the paper it would be good if the authors discussed the multiplexing feature more in detail.

In particular it would be helpful if the authors show more data on the effect of the flow through the long microfluidic channel. In the manuscript the authors mention that some adverse effects can be expected for long channels, related to the pressure with which the fluid needs to be flowed through. This will be the most interesting part for the readership from a technical point of view, and should be elaborated on.

Response to Reviewers

We thank all the reviewers for their valuable comments and suggestions. Please find our point-by-point answer to each reviewer's concern. Changes made in the main manuscript are marked in red font and between quotation marks.

Reviewer #1:

Query 1: Is there any risk of spillover of signal among the 9 samples? This can be checked by correlation of expression profiles of the vertical and horizontal barcodes across the 9 samples. Providing some diagnostic plots along these lines would be very insightful. It may be desirable to design a benchmarking experiment where one can assess the proportion of incorrectly assigned reads to samples or tissue areas.

Answer: We thank Reviewer#1 for the comments and improved our manuscript adding new experiments and analysis to assess the multiplexing function of xDBiT. Therefore, we have performed two new additional experiments. The first experiment addresses the potential cross-contamination occurring during the starting reverse transcription (RevT) reaction of the xDBiT workflow; the second experiment evaluates potential cross-contamination downstream the the RevT reaction.

In the first experiment, we have clamped the 9-well adaptor to the glass substrate and filled the wells with different food dye-colored water. Within 24 hours, we could not detect leakage between the 9-wells. Thus, we conclude that cross-contaminations during the reverse transcription (RevT) step cannot be detected. The result of this experiment has been added to **Supp. Fig. 5E (Figure 1)**:

Figure 1. New supplementary Figure 5E added to the main manuscript.

In a second experiment, we have performed one xDBiT experiment with eight liver sections and tested for cross-contaminations occurring after the RevT step analyzing the corresponding sequencing data. Here, we have found that only 5.5 to 9.5% of the RevT sample barcodes were

cross-contaminations of neighboring samples. Upon double indexing at the RevT and final sample indexing steps, cross-contaminations were fully removed to obtain high quality spatial transcriptome data. The following text has been added to the results and discussion section:

“However, the advantage of sample multiplexing with xDBiT also carried the risk of cross-contamination between samples. To check for potential leakage between the individual wells of the 9-well adapter during the RevT reaction, food dye-colored aqueous solutions were used. Within an interval of 24 h no visible cross-contaminations were detected (**Supp. Fig. 5E**). Subsequently, potential cross-contaminations occurring after the RevT step were investigated in one xDBiT experiment with eight liver sections, leaving the center well of the 3x3 grid empty (**Figure 2E, Methods**). Analysis of the resulting sequencing reads revealed that only 5.5 to 9.5% of the RevT barcodes were cross-contaminations from neighboring samples (**Fig. 2E, and Supp. Table 9**). Notably, from the empty well (Figure 2E, sample, B2) cDNA concentration was not sufficient for a cDNA library preparation. Importantly, with the double barcoding strategy, cross-contaminations can be removed within the *ReadsToCounts* pipeline (see *Methods*).”

“xDBiT allowed barcoding of the samples using the 9-well adapter during the initial RevT reaction and library preparation. We did not observe liquid exchange between the wells of the 9-well adapter. By barcoding samples both at the RevT and the sample indexing level, we demonstrated that less than 9% of the final sequencing reads resulted from cross-contaminations occurring after the RevT step, indicating low cross-contamination among samples. Importantly, within double-barcoded samples, cross-contamination signals can be removed computationally by selecting sequencing reads with matching RevT and indexing sample barcodes. Alternatively, when using only a single barcoding strategy, cross-contamination events can be removed computationally by analysis of the gene expression background. However, we recommend the double-barcoding of samples to exclude the possibility of sample cross-contaminations.”

The cross-contamination experiment has been included in the main text as well as in a new additional illustration of the experiment in **Figure 2E**:

Figure 2. New Figure 2E added to the revised main manuscript. **(E)** Left: Experimental setup of the reverse transcription. Colors denote the barcode that was used for the respective well. Right: Stacked bar plot showing the percentage of reads found in a well, which carried a certain well barcode. Colors denote the barcode and match the experimental outline on the left.

An alternative solution to remove cross-contaminations computationally is to evaluate background signals on spots with no underlying tissue in each sample. We used this approach within our multi-organ experiments:

“The sequencing depth for all organ samples was close to saturation, which was evaluated by computational subsampling analysis (**Supp. Fig. 6**). Samples were only barcoded by indexing primers during the library preparation. For removal of cross-contaminations, the background expression level of genes was measured based on ST spots without underlying tissue. Subsequently, only genes with an expression level higher than twice the standard deviation of the mean background signal were used for downstream analyses. Matching genes of the individual samples before and after background correction against the HOMER database¹ confirmed the depletion of cross-contamination signals (**Supp. Fig. 5B**).”

The results of the filtering have been added as a heat map to **Supp. Fig. 5B**:

Figure 3. New supplementary Figure 5B added to the revised main manuscript.

A detailed description of the benchmarking experiments is added to the method section under the section headlines “Cross-contamination testing in 9-well adapter” and “Analysis of post-RevT cross-contaminations”.

Both methods for removing cross-contaminations are applicable. For further experiments using xDBiT, we recommend using the double barcoding approach. We thank the reviewer for pointing out these important benchmarking experiments and thereby improving the accuracy of xDBiT.

Query 2: Are there any transcript sequences without and x-, y-, or either x-y barcodes in the resulting datasets? i.e. tissues not coming from the PDMS channels? If these transcripts are retained in the final dataset are they used for any further analysis or discarded? Describing this further may clear any confusion for readers.

Answer: A description of how the barcode data was handled has been added to the results section as follows:

“In the first step of the pipeline (*ReadsToCounts*), spatial coordinates and transcript information were extracted from the raw sequencing reads. **Sequencing reads without valid x- or y-barcode were discarded.** After genomic alignment, data were transformed into a spot/gene count matrix.”

In addition, this information has been added to the methods section:

“Unique molecular identifiers (UMIs) and spatial barcodes are extracted from Read 2 using the Drop-seq tool *TagBamWithReadSequenceExtended* and a custom Python 3 pipeline assigns coordinates using barcode information provided in a CSV file. **Reads without a valid x- or y-barcode are discarded in this step.** The *DigitalExpression* function is used to collapse the UMIs and generate a spot-count matrix.”

Query 3a: The abstract states 18 spatially resolved transcriptomics datasets were captured, which suggests to me that 2 9-tissue xDbit experiments were performed.

Answer: We measured in total 18 tissue sections using the xDBiT approach. However, the xDBiT datasets generated from pancreas sections showed only very low quality sequencing data and were discarded for the downstream analysis. The read and gene counts of the pancreas samples have been now added to **Supp. Fig. 5F**:

Figure 4. New supplementary Figure 5F added to the revised main manuscript.

In the new manuscript, we have detailed the number of successfully measured ST data sets in the abstract to 16 and described the pancreas data set in the results:

“To demonstrate the broad applicability of xDBiT, we generated 18 spatially resolved datasets from six different murine organs, including the kidney, heart, cerebellum, spleen, liver, and pancreas (**Fig. 3**). The UMIs and genes per xDBiT spot varied for the organs between 5,000–20,000 and 1,000–5,000, respectively (**Fig. 3A and B**). The pancreas samples showed low UMI and gene counts and were therefore excluded from further analysis (**Supp. Fig. 5F**).

Query 3b: The authors should assess any systematic biases detected among these.

Answer: In Figure 3F, we showed the correlation of expression between the samples, and we have further assessed any systematic biases between the experiments, following the review’s suggestion, using UMAP plots for possible batch effects and added a new **Supp. Fig. 7**:

Figure 5. New supplementary Figure 7.

With the new analysis, we have not detected any systematic biases, and have added the following sentence to the results section:

“After preprocessing and dimensionality reduction using Uniform Manifold Approximation and Projection (UMAP)², the data showed no visible batch effects (**Supp. Fig. 7**). Clustering using the *Leiden* algorithm³ and projection of xDBiT spots onto the respective microscopy images displayed spatially distinct clusters (**Fig. 3C and D**).“

Query 3c: Figure 3F does display the concordance of expression between the samples, however it's not immediately clear which experiment (same or different) the samples came from.

Answer: We appreciate the reviewer’s concern and we have addressed this issue with an additional supplementary Table 7. To make it clear which samples were used for the different experiments, we have added an overview of all samples used in this study to **Supp. Table 7**. The table also contains the respective sample ID used in the course of this manuscript:

Table 1. New supplementary Table 7. **Information about the origin of the samples used in this study.** Every xDBiT run ID refers to one experiment in which one xDBiT PDMS chip was used. The well positions refer to the position on the 3 x 3 grid on a PDMS chip.

Experiment	Organ	Sample ID	xDBiT run ID	Well positions	Mouse ID
Multi-organ	Heart	h1	1	A3	TS26
	Heart	h2	1	B1	TS26
	Kidney	k1	1	B3	TS26
	Kidney	k2	1	C1	TS26
	Liver	l1	1	B2	TS26
	Spleen	s1	1	A1	TS26
	Spleen	s2	1	A2	TS26
	Pancreas	p1	1	C2	TS27
	Pancreas	p2	1	C3	TS27
	Heart	h3	2	A3	TS27
	Heart	h4	2	B1	TS27
	Kidney	k3	2	B3	TS27
	Kidney	k4	2	C1	TS27
	Liver	l2	2	B2	TS27
	Spleen	s3	2	A1	TS27
	Spleen	s4	2	A2	TS27
	Cerebellum	c1	2	C2	NaCai1
	Cerebellum	c2	2	C3	NaCai1
Spillover analysis	Liver	l3	3	A1	TS16
	Liver	l4	3	A2	TS16
	Liver	l5	3	A3	TS16
	Liver	l6	3	B1	TS16
	Liver	l7	3	B3	TS5
	Liver	l8	3	C1	TS5
	Liver	l9	3	C2	TS5
	Liver	l10	3	C3	TS5

The Sample IDs in **Figure 3F** have been updated accordingly:

Figure 6. Figure 3F has been updated in the revised main manuscript.

Query 4: In Figure 4c we observe some horizontal and vertical stripe artifacts. This suggests that some of the channels in the PDMS chip may not work all the time. I would suggest discussing this potential limitation as well as discussing potential ways to address this, either experimentally or algorithmically.

Answer: Stripe artifacts occur due to reagent flow problems in single microchannels during the xDBiT workflow. In consequence differences in the read counts between the channels occurred. These effects were reported in the original DBiT-seq publication⁴. Importantly, smaller differences in the read counts between channels are only visible in the raw count data and can be effectively removed by standard normalization methods. To demonstrate this, we have added a new **Supp. Fig. 4:**

Figure 7. New supplementary Figure 4

The following text was added to the manuscript to explain this type of stripe artifact and describe the results of the normalization:

“To demonstrate the quality of the spatial transcriptomic data, we projected the raw sequencing read counts per spot onto the nuclei images as shown exemplarily for *Actb* in **Supp. Fig. 4A**. Resulting overlay images showed stripe artifacts consisting of rows or columns of spots with higher or lower read counts compared to their neighboring elements. These artifacts have been reported previously⁴ and can be effectively removed by normalizing each spot by the total number of reads of the respective spot (**Supp. Fig. 4B**).”

In the first submitted manuscript, we decided to extract only high quality spot transcriptomes for downstream analysis. Therefore, we used stringent filtering conditions in the preprocessing steps, which led to empty data columns or rows, as pointed out by reviewer #1. The origin and possible solutions to the problem are discussed in the revised version of the manuscript:

“Further downstream analysis showed that stringent read count filtering leads to high-quality data. We show that stripe artifacts, which are visible on the raw count data, can be effectively removed using standard normalization methods⁵. However, the high stringency led in some ST images to the removal of whole column or row elements. Both,

the so-called stripe artifacts and the empty rows or columns, result most likely from non-homogenous flow conditions within the horizontal or vertical channels of the PDMS chip. This could be further optimized by introducing standard fluid interfaces to the chip, ensuring homogenous fluid flow, and handling of air bubbles through micromechanical features in addition to the already implemented bubble traps at the transition from inlet to channel.”

An implementation of the suggested optimization is beyond the scope of this study.

Query 5: I suggest the authors perform the MEFISTO analysis presented in Figure 4 separately per tissue. It's unclear to me why Factor 1 would be shared with brain, spleen and to some extent kidney. It may be more insightful examining the factors determined from each tissue.

Answer: We have followed the reviewers' suggestion and have performed the analysis separately per tissue. To add clarity and understanding in our new manuscript, we have explained the analysis in more detail in the results section:

“For spatial gene expression pattern analysis of the xDBiT ST data, we applied MEFISTO, a factor analysis method to identify the driving sources of gene variation in high-dimensional datasets and account for spatial dependencies⁶. **The factor analysis was performed separately for each tissue section and identified a potentially smaller set of previously unobserved variables, called factors. These factors reveal the covariance structure of the spatial transcriptomic dataset of the respective tissue section.** Sections from the same organ showed a comparable number of factors that explained spatial gene expression variations.”

We have also revised the method section and add additional clarifications:

“MEFISTO factor analysis⁶ was performed using the Python package *mofapy2* (v0.6.4). **Datasets of each tissue section were analyzed separately.** Spatial spot coordinates were used as covariates and only highly variable genes were selected for the analysis.”

Query 6: How applicable is xDbit to archival tissues, e.g. those preserved using FFPE technique?

Answer: We appreciate this timely relevant question. In fact, this question has been previously addressed by the inventors of the original DBiT-Seq method. In short *deterministic barcoding* is compatible with FFPE tissue⁷. We added this information to the discussion of the revised manuscript:

“Deterministic barcoding can also be performed with archived formalin-fixed and paraffin-embedded (FFPE) samples, however with lower read depth⁷. This is expected due to the fact that FFPE-derived RNA is highly degraded and chemically modified, and affects downstream sequencing processes^{8,9}.“

However, RNA quality in FFPE samples depends highly on the storage conditions and storage time. In the course of this project, we have also tested xDBiT on FFPE samples and have found very low cDNA quality in the samples as shown in the Figure below:

Figure 8. Figure for Reviewer 1. Bioanalyzer trace of cDNA retrieved from xDBiT experiment with fresh-frozen sample (left) and xDBiT experiment with FFPE sample (right).

The Figure 8 summarizes that the RNA quality in sections from fresh-frozen kidney samples is far higher than in FFPE kidney samples. Since the goal of this study was the development and benchmarking of xDBiT, we decided to use only fresh-frozen tissue samples. This allowed us to benchmark our method against published datasets and be independent of experimental factors such as storage time and temperature.

Query 7: How applicable is xDbit to histopathological assessment before or after xDbit workflow? e.g. H&E staining and imaging prior to the immunofluorescence staining described.

Answer: xDBiT is compatible with any microscopic readout method that does not interfere with the RNA integrity of the sample. Standard histopathological stainings such as H&E stainings, do not interfere with the RNA quality and were also shown to be compatible in the original DBiT-seq publication⁴. One crucial step for combining different imaging modalities with xDBiT is to register images with the alignment image containing the outer grid lines of the ST spots. We have solved this problem by using the fluorescence nucleus information in high-quality fluorescence and alignment images. In order to illustrate that the registration step is possible with H&E images, we prepared an H&E stained kidney section, removed the H&E staining, and performed an additional IF staining on the same section. Image registration of both image was possible as shown in **Figure 9**:

Figure 9. Figure for Reviewers 2. Serial H&E and immunofluorescence stainings of one kidney FFPE tissue section. (A) H&E staining of kidney FFPE section. (B) Immunofluorescence staining of kidney FFPE section after removal of H&E staining. Colors denote DAPI (blue) and CD31 (green). (C) Demonstration of registration of H&E color image to DAPI fluorescence image. Registration was performed using the SIFT algorithm and results in an overlap of images.

This new experiment demonstrates that our xDBiT workflow is compatible with H&E stainings. Due to the simplicity of the achievement, and the lack of novelty we only provide this figure for the reviewers.

Query 8: line 426 "Error! Reference source not found."

Answer: We did not notice this error in our first manuscript but we have revised and refreshed the reference link, and we will pay attention to it during the proofreading process.

Reviewer #2 (Remarks to the Author):

Query 9: Since this is the main finding in the paper it would be good if the authors discussed the multiplexing feature more in detail.

Answer: We thank reviewer #2 for the insightful question and followed the suggestion by adding extended information in our revised manuscript. Since query 1 by reviewer #1 requested more benchmarking experiments of the multiplexing capability of xDBiT, in our revised manuscript we have added additional information to clarify the multiplexing function of xDBiT and address both reviewers' concerns. Please see for this our answers to query 1.

In the revised results section, we have also added additional information about the RevT primers and the barcoding strategy:

“The reverse transcription (RevT) primer carried a hybridization site to ligate the spatial barcodes in the following working steps, and a poly(T) 3'-end to bind to and reverse transcribe all polyadenylated mRNAs (Supp. Fig. 2). **In addition, the RevT primer contained a unique, 6-bp long sequence to barcode the samples during the RevT reaction (Supp. Fig. 2).**”

Together with the new results of the cross-contamination study, we have added a more elaborated discussion of the multiplexing feature and the different strategies that can be implemented on the xDBiT to the revised discussion section:

“xDBiT allowed barcoding of the samples using the 9-well adapter during the initial RevT reaction and library preparation. We did not observe liquid exchange between the wells of the 9-well adapter. By barcoding samples both at the RevT and the sample indexing level, we demonstrated that less than 9% of the final sequencing reads resulted from cross-contaminations occurring after the RevT step, indicating low cross-contamination among samples. Importantly, within double-barcoded samples cross-contamination signals can be removed computationally by selecting sequencing reads with matching RevT and indexing sample barcodes. Alternatively, when using only a single barcoding strategy, cross-contamination events can be removed computationally by analysis of the gene expression background. However, we recommend the double-barcoding of samples to exclude the possibility of sample cross-contaminations.”

Query 10: In particular it would be helpful if the authors show more data on the effect of the flow through the long microfluidic channel. In the manuscript, the authors mention that some adverse effects can be expected for long channels, related to the pressure with which the fluid needs to be flowed through. This will be the most interesting part for the readership from a technical point of view and should be elaborated on.

Answer: We agree with the reviewer that a discussion of this topic would be of great interest to the technical community. Therefore, we have performed a new experiment in which we have measured the flow rate within the individual microchannels on the xDBiT chip. In our new experiment, we used the flow rates from the microchannels with different lengths to predict the maximum possible channel length on the xDBiT platform under the same flow pressure conditions. The following graph of the respective results was added to **Figure 2D** in the revised manuscript:

Figure 10. New Figure 2D added in the revised manuscript.

Further, the following text has been added to the revised results section to describe in more details the new experiment:

“Sample multiplexing within the xDBiT approach was achieved by the implementation of a serpentine channel design. For this, the microfluidic channels were elongated and the lengths of the resulting channels varied between 117.7 mm and 165.7 mm. We characterized the effect of the channel length on the fluid flow behavior on a PDMS chip by measuring the volumetric flow rate in all 38 channels when applying a constant vacuum of 300 mbar to the outlets. Flow rates showed a negative linear correlation with the channel length as it was expected from the Hagen-Poiseuille equation¹⁰ (**Fig. 2D**). Between the shortest and the longest channels on the PDMS chips the flow rate differed by 26.5%. Time intervals for washing steps were adjusted to the lowest flow rate on the chip to ensure a minimal volume exchange of 15 μL per channel.”

In the discussion section of the revised manuscript, we have also addressed the future implications of these findings and the limitations of the platform:

“One limiting factor of the xDBiT approach is the fluid resistance, which scales linearly with the channel length. From our correlation analysis between the fluid flow rates and microchannel lengths in the xDBiT PDMS chips (**Figure 2D**), we can conclude that microchannels with < 260 mm length can be operated under the chosen pressure conditions. Thus, we anticipate that a higher degree of multiplexing than presented here could be achieved. Longer channels would require, however, a higher fluid forward pressure to drive fluid flow, which in turn would induce leakage between the microchannels and disruption of the underlying tissue.”

Further, we have added the method to reproduce this new experiment in the revised manuscript under the section heading “Flow rate measurement”.

References

1. Zhang, F. & Chen, J. Y. HOMER: a human organ-specific molecular electronic repository. *BMC Bioinformatics* **12**, S4 (2011).
2. McInnes, L., Healy, J. & Melville, J. UMAP: Uniform Manifold Approximation and Projection for Dimension Reduction. (2018).
3. Traag, V. A., Waltman, L. & van Eck, N. J. From Louvain to Leiden: guaranteeing well-connected communities. *Sci. Rep.* **9**, 5233 (2019).
4. Liu, Y. *et al.* High-Spatial-Resolution Multi-Omics Sequencing via Deterministic Barcoding in Tissue. *Cell* **183**, 1665-1681.e18 (2020).
5. Luecken, M. D. & Theis, F. J. Current best practices in single-cell RNA-seq analysis: a tutorial. *Mol. Syst. Biol.* **15**, e8746 (2019).
6. Velten, B. *et al.* Identifying temporal and spatial patterns of variation from multimodal data using MEFISTO. *Nat. Methods* 1–8 (2022) doi:10.1038/s41592-021-01343-9.
7. Liu, Y., Enniful, A., Deng, Y. & Fan, R. Spatial transcriptome sequencing of FFPE tissues at the cellular level. *bioRxiv* 2020.10.13.338475 (2020) doi:10.1101/2020.10.13.338475.
8. Ahlfen, S. von, Missel, A., Bendrat, K. & Schlumpberger, M. Determinants of RNA Quality from FFPE Samples. *PLOS ONE* **2**, e1261 (2007).
9. Groelz, D. *et al.* Non-formalin fixative versus formalin-fixed tissue: A comparison of histology and RNA quality. *Exp. Mol. Pathol.* **94**, 188–194 (2013).
10. Oh, K. W., Lee, K., Ahn, B. & Furlani, E. P. Design of pressure-driven microfluidic networks using electric circuit analogy. *Lab. Chip* **12**, 515–545 (2012).

REVIEWERS' COMMENTS

Reviewer #1 (Remarks to the Author):

I thank the authors for their careful addressing of my and other reviewers' comments. The addition of the clamping experiment and examination of potential cross-contamination along the serpentine channels adds significant confidence to the xDBiT approach and shows any effects are small. In addition, the removal of horizontal and vertical stripe patterns with standard normalisation also points to broad applicability of xDBiT. The explanation of sample applicability to archival tissues and histopathological assessment, although covered in the previous publication, is also appreciated.

Reviewer #2 (Remarks to the Author):

The authors have satisfactorily answered my comments.

Response to Reviewers

We hereby want to thank the reviewers for their productive feedback on our manuscript and are pleased to hear that our revisions satisfactorily answered all questions.

Reviewer #1 (Remarks to the Author):

I thank the authors for their careful addressing of my and other reviewers' comments. The addition of the clamping experiment and examination of potential cross-contamination along the serpentine channels adds significant confidence to the xDBiT approach and shows any effects are small. In addition, the removal of horizontal and vertical stripe patterns with standard normalisation also points to broad applicability of xDBiT. The explanation of sample applicability to archival tissues and histopathological assessment, although covered in the previous publication, is also appreciated.

Reviewer #2 (Remarks to the Author):

The authors have satisfactorily answered my comments.